



# Ozone Profile Retrieval from nadir TROPOMI measurements in the UV range

Nora Mettig[1], Mark Weber[1], Alexei Rozanov[1], Carlo Arosio[1], John P. Burrows[1], Pepijn Veefkind[2], Anne M. Thompson[3], Richard Querel[4], Thierry Leblanc[5], Sophie Godin-Beekmann[6], Rigel Kivi[7], and Matthew B. Tully[8]

[1]Institute of Enivronmental Physics, University of Bremen, Bremen, Germany
[2]Royal Netherlands Meteorological Institute (KNMI), De Bilt, Netherlands
[3]NASA/Goddard Space Flight Center, Greenbelt, MD, USA
[4]National Institute of Water & Atmospheric Research (NIWA), Lauder, New Zealand
[5]Jet Propulsion Laboratory, California Institute of Technology, Wrightwood, CA, USA
[6]LATMOS, Sorbonne University, Paris, France
[7]Finnish Meteorological Institute, Sodankylä, Finland
[8]Bureau of Meteorology, Melbourne, Australia

**Correspondence:** Nora Mettig (mettig@iup.physik.uni-bremen.de)

**Abstract.** The TOPAS algorithm to retrieve vertical profiles of ozone from space-borne observations in nadir viewing geometry has been developed at the Institute of Environmental Physics (IUP) of the University of Bremen and applied to TROPOMI L1B spectral data version 2. The spectral data between 270 and 329 nm are used for the retrieval. A re-calibration of the measured radiances is done using ozone profiles from MLS/Aura. Studies with synthetic spectra show that individual profiles in the stratosphere can be retrieved with the accuracy of about 10%. In the troposphere, the retrieval errors are larger depending on the a-priori profile used. The vertical resolution above 18 km is about 6 – 10 km and it degrades to 15 – 25 km below. The vertical resolution in the troposphere is strongly dependent on the solar zenith angle (SZA). The ozone profiles retrieved from TROPOMI with the TOPAS algorithm were validated using data from ozone sondes and stratospheric ozone lidars. Above 18 km, the comparison with sondes shows excellent agreement within less than ± 5% for all latitudes. The standard deviation of mean differences is about 10%. Below 18 km, the relative mean deviation in the tropics and northern latitudes is still quite good remaining within ± 20%. At southern latitudes larger differences of up to +40% occur between 10 and 15 km. The standard deviation is about 50% between 7-18 km and about 25% below 7 km. The validation of stratospheric ozone profiles with ground-based lidar measurements also shows very good agreement. The relative mean deviation is below ± 5% between 18 – 45 km with a standard deviation of 10%. TOPAS retrieval results for one day of TROPOMI observations were compared to MLS and OMPS-LP data. The relative mean difference was found to be largely below ±5% between 20 – 50 km with exception of very high latitudes.



## 1 Introduction

Ozone is one of the most important trace gases in the Earth's atmosphere. The stratospheric ozone layer is of particular

importance for humans because it protects the biosphere from biologically damaging ultraviolet radiation (UV). Ozone is a toxic gas and a strong oxidant. Consequently in the troposphere it impacts negatively on human health, ecosystem services and agriculture. Furthermore, tropospheric ozone is a potent greenhouse gas. Ozone also plays an important role in many aspects of atmospheric chemistry, physics, and dynamics. In the stratosphere, ozone is mainly determined by the Chapman cycle (Chapman, 1930) and catalytic loss cycles (Bates and Nicolet, 1950; Salawitch, 2019). The ozone layer heats the stratosphere

and thus leads to a vertical temperature maximum ("inversion layer"). The cooling of the stratosphere by ozone depletion and due to climate change is still a subject of research today. Ozone received much public attention when in the 1980s, its enormous reduction was observed during Antarctic spring (Farman et al., 1985). This so-called ozone hole is the result of human-made chlorofluorocarbon compounds and is still part of scientific research today. The recovery of the ozone hole is monitored continuously (World Meteorological Organization, 2018). In order to separate dynamical and chemical effects on

ozone that are varying with altitude, it is not sufficient to limit measurements to total ozone column amounts. The key for ozone monitoring is the precise measurement of its vertical distribution at high spatial and temporal resolution. Two common and accurate methods to measure ozone profiles are ozone sondes and lidars. However, neither technique can provide sufficient spatial and temporal coverage. For this reason, the determination of a global ozone distribution is only possible from satellite data.

In addition to the nadir-viewing geometry used here, the atmosphere can also be scanned using limb- and solar (lunar/stellar) occultation viewing. This technique has been used among others by OSIRIS ( launched 2001, Llewellyn et al. (1997)), SCIA-MACHY (launched 2002, Burrows et al. (1995)), MLS (launched 2004, Waters et al. (2006)), SAGE III (launched 2001 in Meteor-3M and 2006 on ISS, Cisewski et al. (2014)) and OMPS (launched 2011, Flynn et al. (2014)). Ozone profiles from limb and occultation data generally have higher accuracy and a higher vertical resolution than nadir-viewing instruments (Has-

sler et al., 2014). However, the typical along-track spatial resolution is much higher for nadir instruments (6 km to 200 km). There is also the risk of a gap in limb observations in the future, since only very few new limb missions are in planning, while observation programs using nadir-viewing satellites extend well into the 2030s.

In 1957 a first theoretical retrieval of vertical profiles of ozone from space by using passive remote sounding in the ultraviolet (Singer and Wentworth, 1957) was described. UV radiation has a lower penetration depth into the atmosphere at shorter

wavelengths, thus permitting the retrieval of vertical ozone profiles using radiation back-scattered from the atmosphere and measured by nadir-viewing satellite instruments. With the launch of NIMBUS 4 in 1970 and the back-scatter ultraviolet (BUV) spectrometer instrument on it, the measurement of vertically resolved ozone in the atmosphere from space became possible. Regular and continued daily observations started with the solar back-scatter ultraviolet instruments SBUV (1978) and SBUV/2 (since 1985). The SBUV ozone profile retrieval uses up to twelve discrete wavelengths (Bhartia et al., 1996). Beside nadir ozone

retrieval using the ultraviolet spectral region, the infrared range is also convenient. Nadir sounding in the thermal infrared has a high sensitivity for O3 in the upper troposphere and lower stratospheric (Turquety et al., 2002).





The first measurements at the top of the atmosphere of contiguous spectra in the ultraviolet and visible spectral regions at sufficiently high spectral resolution were made by the the Global Ozone Monitoring Experiment (GOME) instrument (Burrows et al., 1999). These data were used to retrieve ozone profiles from measurements of GOME (Burrows et al., 1999). Munro et al.

(1998) demonstrated that the ozone profiles retrieved by using the Optimal Estimation technique were particularly sensitive to the lower stratosphere and the troposphere. Hoogen et al. (1999) highlighted the need for absolute calibration of the satellite measurements to obtain reliable results from GOME. Hasekamp and Landgraf (2001) chose a slightly different retrieval approach by using a Tikhonov regularisation, which employs smoothing constraints, instead of the Optimal Estimation method, that relies on a-priori constraints from an ozone profile climatology. van der A (2002) re-calibrated the measured spectra from

GOME in order to further improve the GOME ozone profiles. Liu et al. (2005) showed that with an extensive spectral correction and a correction of the wavelength scale information on tropospheric ozone can be further improved.

With the subsequent instruments SCIAMACHY (2002), OMI (2004) and GOME-2 (2006) the spatial resolution of the measurements was improved. Using the OPERA algorithm, van Peet et al. (2013) were able to determine ozone profiles using data from multiple instruments (GOME and GOME-2) with identical settings to obtain for the first time a merged time series,

which required a subsequent correction of calibration offsets and a correction for degradation. Miles et al. (2015) developed a retrieval algorithm for GOME-2, which consists of three steps and aims at an even more accurate determination of tropospheric ozone. In the first step, the stratospheric profile is determined from shorter wavelengths of the Hartley ozone absorption band (266 nm – 307 nm), in the second step the surface albedo is retrieved using the radiance at 336 nm, and in the last step, the tropospheric profile is retrieved only from the Huggins ozone band (323 – 335 nm). For OMI, Liu et al. (2010) showed that

ozone retrieval with good accuracy (up to 10% in the troposphere) is possible after a spectral re-calibration of the Level 1 data.

With the launch of Sentinel 5 Precursor (S5P) in October 2017, TROPOMI is another nadir-viewing spectrometer operating in the UV/Vis and SWIR spectral range. It is a follow-up of OMI and SCIAMACHY. Using TROPOMI data, it is possible to continue time series of past and current instruments into the future. The particular advantage of TROPOMI is its unprecedented spatial resolution of $28.8 \times 5.6$ km$^2$ in the UV band 1. The main objective of this study is the first evaluation of UV radiance

data from TROPOMI for the ozone profile retrieval with the TOPAS (Tikhonov regularized Ozone Profile retrievAl with SCIATRAN) approach, which is the successor of FURM (Full Retrieval Method) algorithm (Hoogen et al., 1999). The latest pre-launch analyses showed that a re-calibration was necessary, especially in the UV range, which leads to an improved level 1B (L1B) spectral data version to Version 2 (Ludewig et al., 2020). Since the determination of ozone profiles requires even higher absolute accuracy of the spectra, additional steps in the calibration correction were needed and are presented in this

paper.

In the future, an operational TROPOMI Ozone Profile L2 product will also be provided. Due to the on-going re-calibration, this is currently (end of 2020) delayed and is expected to be released in summer 2021. So far the L1B version 1 TROPOMI data of band 3 (314 – 340 nm) have been used by Zhao et al. (2020) to determine tropospheric ozone and investigate its changing distribution due to the Covid-19 pandemy.

This paper is structured as follows. Section 2 describes the data used in this paper. The TOPAS retrieval method is described in Section 3, and in Section 4, the retrieval quality based upon a sensitivity study using synthetic spectra is provided. Section



5 discuss the additional implemented calibration correction. In Section 6, the TOPAS ozone profile retrieval is validated with ozone sondes and lidar measurements. First results and a comparison to MLS and OMPS limb measurements are shown in Section 7. Finally, a summary is given in Section 8.

## 2 Measurement data

Beside TROPOMI measurements, which are used to retrieve ozone profiles, data sets from other instruments are used in this study for calibration and validation purposes. In particular, ozone profiles from collocated MLS measurements are used to derive calibration corrections for TROPOMI radiances. In addition, these profiles are exploited, together with OMPS-LP data, for an initial verification of the ozone profiles retrieved from TROPOMI. A more extensive validation is performed using 95 globally distributed ozone sonde and lidar measurements.

### 2.1 TROPOMI

TROPOMI (TROPOspheric Monitoring Instrument) is a nadir viewing spectrometer and the only instrument on the Sentinel-5 Precursor (S5P) satellite launched in October 2017. S5P is part of the Copernicus Earth observation programme and is designed to prevent a potential gap in global atmospheric monitoring that could arise between existing missions such as OMI 100 and GOME-2 and the upcoming Sentinel-4 and Sentinel-5 (Fletcher and McMullan, 2016). S5P is in a sun-synchronous orbit with an equator crossing time of 13:30 local time. TROPOMI consists of four spectrometers in the UV, UVIS, NIR and SWIR spectral range. For the ozone profile retrieval, UV1 and UV2 bands are used. Both are located on the same CCD detector. The wavelength range of UV1 extends from 267 to 300 nm and that of UV2 from 300 to 332 nm. The spectral resolution of both bands is 0.5 nm, and the sampling is 0.065 nm. The high spatial resolution of TROPOMI is worth mentioning. One 105 measurement pixel in the middle of the swath covers $28.8 \times 5.6$ km$^2$ (cross-$\times$along-track) in UV 1 and $3.6 \times 5.6$ km$^2$ in UV2. The difference between the two channels comes from the different binning factors that are already applied on-board. As the intensity of the measured radiation below 300 nm is much lower, the detector pixels in UV1 have to be binned to get an adequate signal-to-noise ratio. For the ozone profile retrieval, the sampling of UV1 and UV2 has to be matched. To further improve SNR in both bands, additional binning is applied before the retrieval.

In the middle of the swath one UV1 pixel and eight UV2 pixels are binned in across-track direction while in the along-track direction 8 pixels are binned for both UV1 and UV2. At the far end of the swath the UV1 pixels have half size in the across-track direction and therefore 2 UV1 pixels have to be binned here to obtain the same size of the eight UV2 pixels. That results in a spatial sampling of about $45 \times 45$ km$^2$ for all pixels. This additional binning reduces the computation time per satellite orbit.

115 In this study, we use the L1B product of an updated processor version (version 2.0), which is available from the end of 2019. Details of the pre-launch calibration and of the V2 update can be found in Ludewig et al. (2020). Only the V2 update is of sufficient quality for the ozone profile retrieval. At the moment, only a limited data set of version 2.0 is available. The data are not yet officially released. Modifications in V2 data are still possible until the official release. Between June 2018 and October





2019 data from 2 weeks every three months are processed (all in all around 12 full weeks). TROPOMI, like all instruments
of this type, also shows drift and degradation effects in the UV channels. Since the same optical path is used for measuring
radiance and solar irradiance, these effects cancel to a large degree in the retrieval using sun-normalised radiances. Remaining
uncertainties due to errors or changes in the absolute radiometric calibration need to be corrected for. This calibration correction
was performed for UV1 and UV2 bands by comparisons with OMPS solar data. The measured TROPOMI irradiance in the
wavelength range between 270 nm and 332 nm is between 5 and 15% lower and was corrected by these values in version 2 of
the L1B product (Ludewig et al., 2020).

For the ozone profile retrieval, all ground pixels are used which do not have an error flag above 15 for "measurement_quality"
or a "ground_pixel_quality" flag greater than 32. Besides the measured radiance and irradiance, the signal-to-noise ratios
contained in the L1B product are used. Here, only single pixels which have a mean SNR(UV1) > 20 or SNR(UV2) > 50
are accepted. These low SNR limits of the single pixels are then increased by binning (with $n$ pixels) as described above. A
Gaussian approach is used for the increasing binning factor: $SNR_{binned} = 1/\sqrt{2n} \cdot \sum_{i=1}^{n} SNR_i$.

## 2.2 MLS

MLS is a forward-looking microwave limb sounder aboard Aura that was launched in July 2004. It measures millimetre and
submillimetre emissions by seven radiometers which cover a spectral region between 118 GHz and 2.5 THz (Waters et al.,
2006). Aura is moving on a sun-synchronous orbit with an equatorial passing time of 13:45 local time. MLS has a spatial
sampling of ~6 km across-track and ~200 km along-track. Although TROPOMI and MLS do not operate on the same satellite,
their measurements are quite close. The maximum distance between the closest MLS and TROPOMI pixels is 1000 km and
1.5 hours.

MLS is an extensively characterised and validated instrument measuring among others vertical ozone profiles (Froidevaux
et al., 2008; Livesey et al., 2008). The temporal stability was also shown by comparison with, e.g. lidar measurements (Nair
et al., 2012). The approved vertical range in which the ozone profiles may be used is between 9 km and 75 km. The vertical
resolution varies from 2.5 to 3.5 km from the upper troposphere to the middle mesosphere. The precision is estimated to be
5-100% at 9-20 km, 2-4% at 20-45 km, and 7-30% at 45-60 km. The accuracy varies from 7% to 10% in the troposphere and
is around 5% in the stratosphere. Version 4.23, which we use, differs very little from the previous versions, especially in the
stratosphere.

## 145 2.3 OMPS

Another data set suitable for comparison with TOPAS ozone profiles is provided by the Ozone Mapping and Profiler Suite
(OMPS) that was launched on board of the Soumi National Polar-Orbiting Partnership (SNPP) at the end of 2011. SNPP has
a sun-synchronous orbit with the ascending node at 13:30 local time. That means TROPOMI and OMPS operate in a loose
formation at a distance of less than 5 minutes. The measurements used for the ozone profiles retrieval are taken from the limb
profiler (LP) and are available since 2012 (Flynn et al., 2014).





We compare the ozone profiles from TOPAS retrieval with the OMPS-LP profiles retrieved at IUP Bremen by Arosio et al. (2018). The vertical resolution of the OMPS-LP ozone profiles varies from 1.5 to 4.5 km. The retrieval error resulting from the measurement noise is 1-4% above 25 km increasing up to 10-30% in the upper troposphere. The accuracy varies from 5-10% in the whole altitude range, except in the lower tropical stratosphere where a bias of 10% with respect to ozone sondes is observed.

## 2.4 Ozone sondes

Balloon-borne ozone sondes provide a well-established dataset of in-situ measured ozone profiles in the troposphere and lower stratosphere. Ozone sondes can be operated up to an altitude of approximately 35 km and have a vertical resolution of 100-150 m depending on the design and environmental conditions. The measurement precision is 3-5%, and the accuracy is 5-10% (Deshler et al., 2008; Johnson, 2002; Smit et al., 2007). Ozone sondes have been used in many studies on tropospheric and stratospheric ozone and especially for the validation of satellite data (e.g. Kroon et al., 2011; Jia et al., 2015; Huang et al., 2017; Hubert et al., 2020).

The ozone sondes data for the validation come from the World Ozone and Ultraviolet Radiation Data Center (WOUDC) (WOUDC Ozonesonde Monitoring Community et al.) and the Southern Hemisphere Additional Ozonesondes (SHADOZ) (Witte et al., 2017, 2018; Thompson et al., 2017; Sterling et al., 2017). During the validation period (June 2018 to October 2019) data from 26 WOUDC and 9 SHADOZ stations were available. The stations are displayed in the map in Fig. 1. The collocation criteria for comparison with TROPOMI are 100 km maximum distance and 24 hours time difference. SHADOZ profiles were filtered to exclude data that displayed 'dropoffs' larger than 5% (Stauffer et al., 2020). In total, 231 WOUDC and 22 SHADOZ sonde profiles were compared with TROPOMI ozone profiles.

## 2.5 Ozone lidar

For the validation of ozone profiles, particularly in the upper stratosphere, stratospheric lidar measurements are recommended. The high power differential absorption lidars (or DIAL) are designed for precise measurements of stratospheric ozone concentration from 20 to 50 km altitude. They use two or more lasers wavelengths with strong and weak ozone absorption to measure the backscattered radiation and determine the ozone concentration in the atmospheric layer from their difference. In general, the estimated accuracy of the lidar ozone profiles is 5% in the 15 – 50 km range (Leblanc et al., 2016). Like the ozone sondes, lidar measurements are regularly used to validate satellite ozone profiles (e.g. Rozanov et al., 2007; Jiang et al., 2007; Hubert et al., 2016).

For our validation, we used measurements from five lidar stations that are part of the NDACC network (de Mazière et al., 2018). They are marked green in Fig. 1 : Hohenpeissenberg (Germany), Lauder (New Zealand), Mauna Loa (Hawaii), Table Mountain (California), and Observatoire de Haute Provence (France). For lidar measurements, we used the same matching criteria as for ozone sondes and found a total of 177 matching measurements during the validation period.



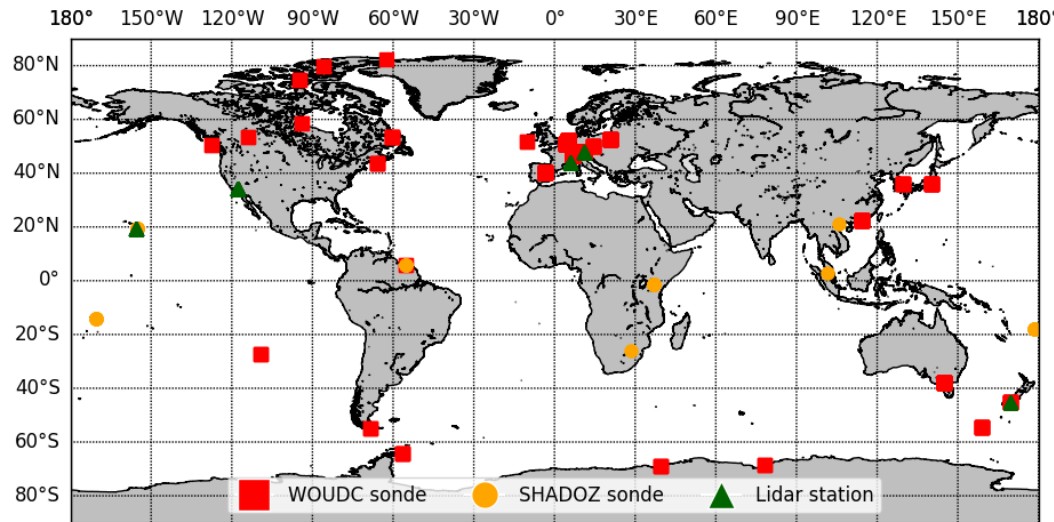

**Figure 1.** Global distribution of ozone sondes (WOUDC and SHADOZ) and lidar measurements used for comparison with the TROPOMI ozone profiles. The exact number and position can be found in Table S1 (supplement).

## 3 Retrieval method

### 3.1 Inversion technique

185 The IUP Bremen retrieval algorithm, which will be referred to as TOPAS (Tikhonov regularized Ozone Profile retrievAl with SCIATRAN), is based on the first-order Tikhonov regularisation approach (Tikhonov, 1963) as described in detail by Rodgers (2002). The relationship between the state vector $x$ that contains the atmospheric quantities to be retrieved and the measurement vector $y$ is given by the forward model operator $F(x)$: $y = F(x) + \epsilon$, where $\epsilon$ represents all errors. For the linearisation of the problem the derivative $K$ of the forward model is needed, that is called the Jacobian or weighting function matrix:

$$K = \frac{\partial F(x)}{\partial x}. \tag{1}$$

190 The linearised expression with the first guess state vector $x_0$ is then:

$$y = F(x_0) + K(x - x_0) + \epsilon. \tag{2}$$

The inverse problem is ill-posed and can be solved by minimising the following norm:

$$\|(y - F(x_0) - K(x - x_a) - \epsilon)\|_{S_y^{-1}} + \|(x - x_a)\|_{S_r} \to \min. \tag{3}$$

Here $S_y$ is the measurement error co-variance matrix and $S_r$ is the regularisation matrix. The latter contains contributions from
195 both the 0-th and 1-st order Tikhonov terms and given by:

$$S_r = (S_a^{-1} + \gamma S_t)^T (S_a^{-1} + \gamma S_t) \tag{4}$$


where the zeroth order Tikhonov term is represented by the a-priori covariance matrix, $S_a^{-1}$ (see e.g. Rodgers (2002)), $S_t$ is the first order derivative matrix (see e.g. (Rozanov et al., 2011)) and $\gamma$ a scaling factor.

Since the problem is not linear, an iterative approach is necessary. Here, a Gauss-Newton iteration scheme is used as described in detail by Rodgers (2002). It should be noted, that in the first iteration $x_0 = x_a$. In the subsequent iterations $x_0$ is replaced by the solution obtained from the previous iteration, while $x_a$ remains fixed. The solution at iteration step $i+1$ is given by:

$$x_{i+1} = x_a + [K^T S_y^{-1} K + S_r]^{-1} [K^T S_y^{-1} (y - F(x_i) - S_r(x_i - x_a)]. \tag{5}$$

As in the TOPAS algorithm the relative deviations from the a-priori state are retrieved, the appropriate transformation of the state vectors, Jacobian and regularization matrix is done:

$$\widetilde{x}_{i+1} = X_a^{-1} x_{i+1}, \quad \widetilde{x}_a = X_a^{-1} x_a = 1, \quad \widetilde{K} = K X_a \quad \text{and} \quad \widetilde{S}_r = X_a S_R X_a, \tag{6}$$

where 1 is the unity vector and $X_a = diag\{x_a\}$. Equation (6) is written then as

$$\widetilde{x}_{i+1} = 1 + [\widetilde{K}^T S_y^{-1} \widetilde{K} + \widetilde{S}_r]^{-1} \widetilde{K}^T S_y^{-1} (y - F(x_i) + \widetilde{K}(\widetilde{x}_i - 1). \tag{7}$$

We note that this transformation also affects the retrieval error matrix, $S_e$, and the averaging kernel matrix, $A$. The matrices for the transformed variables are related to those for the standard inversion described by Rodgers (2002) as follows:

$$\widetilde{A} = X_a^{-1} A X_a \quad \text{and} \quad \widetilde{S}_e = X_a^{-1} S_e X_a. \tag{8}$$

## 3.2 Retrieval algorithm

The TOPAS retrieval approach is structured as follows: First, the a-priori information in the first iteration or the results from the previous iteration serve as input to the forward simulations with the radiative transfer model (RTM) in the next iteration. A polarisation correction is applied to the simulated intensities. In a subsequent pre-processing step, the calibration correction spectrum and the rotational Raman scattering are fitted to the radiance spectrum and a 1st order polynomial is subtracted. Finally, the vertical ozone profile and a Lambertian (scalar) surface albedo are retrieved using Eq. 8.

The inversion of the ozone profile is an ill posed mathematical problem. Consequently the retrieval needs to be constrained by a-priori information. Information on total ozone is needed, because it helps to determine the ozone profile selected from the climatology as a-priori.. Pressure and temperature profiles from a reanalysis, and effective scene height are input to the forward model. The a-priori profiles for ozone are taken from the ozone climatology of Lamsal et al. (2004), which contains ozone profiles averaged from sonde and satellite data depending on latitude, season, and total ozone. For each processed profile, the a-priori ozone profile is scaled with an initial value for the total ozone. In order to obtain the best possible starting point, the total ozone and the effective scene albedo from the WFDOAS retrieval (Coldewey-Egbers et al., 2005; Weber et al., 2018) applied to TROPOMI is used. Pressure and temperature profiles are taken from the ERA5 reanalysis (Hersbach et al., 2020) The effective scene height accounts for cloud effects and is calculated using surface height, cloud coverage and cloud-top-height. The cloud



**Table 1.** Overview of the ozone profile retrieval settings.

| Parameter | Setting |
|---|---|
| Radiative transfer model | SCIATRAN V4.1 |
| | pseudo-spherical atmosphere |
| | no polarisation and no rotational Raman scattering |
| Polarisation | not included in RTM, given by LUT |
| Rotational Raman scattering | not included in RTM, given by LUT |
| Regularisation | Tikhonov 0th order parameter: 11.11 |
| | Tikhonov 1st order parameter: 0.007 |
| Retrieved quantities | vertical ozone profile |
| | scalar albedo |
| Wavelength range | 270 – 329 nm for ozone profile |
| | 310 – 329 nm for scalar albedo |
| ozone absorption cross-section | Serdyuchenko et al. (2014) |
| ozone profile climatology | Lamsal et al. (2004) |
| Aerosols | no aerosols |
| Vertical grid | 0 - 60 km, 1 km steps |

parameters are part of the operational ESA data product (Loyola et al., 2018). The cloud fraction and cloud-top-height are taken from the offline total ozone S5P product, which contains the OCRA cloud fraction and ROCINN_CRB (cloud as reflecting boundary) cloud altitude matched to the UV3 channel. The cloud-top-height and surface height are weighted according to cloud
coverage to determine the effective scene height (Coldewey-Egbers et al., 2005). That enables cloudy and cloudless pixels to be retrieved without a need to account for clouds implicitly in the RTM.

The radiative transfer model SCIATRAN V4.1 is used for the forward simulations (Rozanov et al., 2011). The radiance in the wavelength range from 270 nm to 329 nm is simulated with the spectral resolution and sampling of TROPOMI. Polarisation and rotational Raman scattering are omitted for reasons of computing time and are accounted for by using a look-up table
(LUT). Instead of a full-spherical atmosphere, a pseudo-spherical atmosphere is assumed, which accelerates the calculation even more. In this approximation, the direct solar beam is calculated for a fully spherical atmosphere, while for the scattered light, a plane-parallel atmosphere is assumed (Rozanov et al., 2000). This might, however, result in larger errors for larger viewing angles. These errors are largely mitigated if the forward model is run by using the angles (viewing, solar zenith and azimuth) at the surface rather than those at the top of the atmosphere (de Beek et al., 2004).
The radiance spectrum calculated by the RTM is corrected for polarisation using a wavelength-dependent scaling factor. To determine this factor, simulated spectra with and without polarisation are taken from the LUT with appropriate values for albedo, total ozone, geometric height, and viewing geometry. These spectra are then convolved with the TROPOMI instrument response function (ISRF). The ratio of both is then used as a correction factor to account for the polarisation effects.





To account for atmospheric and instrumental effects which are not included in the forward modelling, the pre-processing fit

procedure includes three pseudo-absorbers and accounts for a possible misalignment of the wavelength grids of the measured and modelled spectra by performing shift and squeeze correction. During the pre-processing step, the original wavelength range is divided into three spectral windows: 270 – 300 nm (UV1 for TROPOMI), 300 – 310 nm (lower UV2) and 310 – 329 nm (upper UV2). For each of these spectral windows, pseudo-absorbers and shift/squeeze corrections were fitted independently. The first pseudo-absorber used in the fit, which accounts for the missing contribution from the rotational Raman scattering in

the forward model, is the Ring spectrum. The Ring spectrum is obtained from LUT using the same procedure as for the polarisation correction spectrum (ratio of convolved radiances modelled with and without rotational Raman scattering contribution). The second pseudo-absorber represents the calibration correction, which accounts for errors in the stray-light correction and other systematic errors in the radiometric calibration parameters. The calibration correction spectrum is determined using the radiances modelled with ozone information from collocated MLS/Aura measurements as described in details below. The third

pseudo-absober accounts for a wavelength independent offset in the radiance spectra and is represented by the inverse solar spectrum. In addition, a first order polynomial (linear term) is subtracted in the second spectral window (300 – 310 nm).

To cope with nonlinear ill-posed problems, the Tikhonov regularisation has been proved. Especially when a lack of stability occurs, which is indicated, for example, by oscillating ozone profiles, this type of regularisation is proposed. In the key retrieval step the first order Tikhonov regularisation is employed, which ensures that the ozone profile retrieval remains stable and

converges even if the number of independent pieces of information is much lower than the total number of the retrieval grid levels. As the zeroth order Tikhonov term, the inverse a-priori covariance matrix, $S_a^{-1}$ is used. The a-priori variance, which is intended to keep the solution within the natural variability from its a priori value (Rodgers, 2002), is set to 0.3 for both the Lambertian surface albedo and ozone number densities at all altitude levels. The assumed a-priori variance for ozone is in agreement with the findings of (Lamsal et al., 2004), who reported the variability of ozone to be generally less than 30% in

the upper stratosphere and up to 60% in the troposphere. In this study, we preferred to use the altitude independent (constant) a-priori variance for ozone rather than the values from the climatology as the latter might introduce vertical irregularity in the retrieval sensitivity distorting the shape of averaging kernels and occasionally resulting in retrieval artefacts. The strength of the first order Tikhonov term is controlled by the altitude independent scaling factor $\gamma$, see Eq.(4), which is set to 0.007. The optimum value for this scaling factor is unknown, but through empirical studies we found that this value has the largest

information content in the retrieval and the RMS between measurement and forward model is the smallest. The measurement error co-variance matrix, $S_y$, was filled with squared signal-to-noise ratio (SNR) values, which for TROPOMI instrument are reported for each spectral measurement in the level 1B product. The noise is a measure for the one standard deviation random error of the radiance measurement and it is assumed to be spectrally uncorrelated, i.e. all off-diagonal elements of the noise covariance matrix are set to zero.

The state vector comprises the ozone number densities at the retrieval grid levels and the effective Lambertian surface albedo. The vertical grid of the retrieval ranges from the effective scene height to the top of the atmosphere (at 60 km) in steps of 1 km. Within one iterative step, the ozone profiles and the effective surface albedo are retrieved independently, i.e. no cross-talk between these parameters is considered. For the ozone profile retrieval, the wavelength range from 270 to 329 nm is used,





while the albedo retrieval is done using the 310 – 329 nm spectral range. A wavelength-independent (constant) surface albedo

retrieved from the wavelengths above 310 nm is used in the next iterative step for the entire spectral range. This is done because the radiation at shorter wavelengths barely reaches the ground and, thus, the surface albedo for wavelengths below 310 nm is much more challenging to determine. We note that the retrieved effective Lambertian surface albedo is not merely the surface reflection but also includes contribution from the back-scattering of the radiation by aerosols and clouds. The combined use of the effective scene height and effective albedo eliminates the need to include the contributions from the tropospheric aerosols

and clouds in the forward modelling. The iterative process ends when one of the two convergence criteria is fulfilled: change of the ozone number density in a selected height range or change of the spectral fit RMS (the difference between the measured and modelled radiances) from the corresponding values at the previous iterative step is below 2%.

## 4   Retrieval quality

### 4.1   Synthetic retrievals of ozone profiles

One of the widely used approaches to check the self-consistency of the retrieval, investigate its sensitivity and estimate uncertainties and biases in the retrieved parameters is to undertake sensitivity studies. We call the data sets, used in these sensitivity studies, synthetic retrievals of ozone profiles. To this end, the radiances are simulated using the forward model for a representative set of geophysical scenarios. The retrieval algorithm is run then with the simulated radiances instead of the measured ones. The advantage of this approach it that the true state of the atmosphere is known, which is never the case for the real data.

In this study, the radiances are simulated in the spectral range from 270 to 329 nm (TOPOMI UV1 and UV2 bands). The spectral resolution and sampling are set to 0.5 nm and 0.065 nm, respectively, which is in accordance with the specifications of the TROPOMI instrument. All spectra are simulated assuming a fully spherical atmosphere and multiple scattering. The solar irradiance spectrum from Chance and Kurucz (2010) convolved with TROPOMI instrument response function is used in the simulations. To investigate the uncertainty associated with the usage of LUTs to account for the polarisation and the rotational

Raman scattering (see supplement S1: Fig. S1 – S4) the simulated radiances were calculated with the following settings: (i) without both Ring effect and polarisation, (ii) with polarisation only (no Ring effect), and, (iii) including Ring effect only (no polarisation). Furthermore, the effect of the viewing and illumination geometry as well as of the surface albedo on the resulting retrieval uncertainties is investigated. The ozone profiles used to calculate the simulated radiances are taken from the CAMELOT study (Levelt and Veefkind, 2009), whose general objective was to establish the quality requirements for air quality

and climate monitoring by satellites. Three ozone profile scenarios were selected: European background, China polluted, and South polar. Furthermore, two variants of the European ozone profile with enhanced tropospheric pollution were created: one limited to enhancement in the planetary boundary layer and the other enhanced uniformly over the entire troposphere. The pressure and temperature profiles were also taken from the CAMELOT scenarios. In Table 2, a complete overview of the parameters used in this study is given. In total, 1500 synthetic spectra were generated.

The settings for the synthetic retrieval are kept as close as possible to those of the real TROPOMI retrieval. However, a few settings needed to be adjusted. Instead of a measured solar spectrum, we use in the synthetic retrieval the solar spectrum



**Table 2.** Input parameters for the generation of synthetic spectra.

| Parameter | Setting |
|---|---|
| RTM setting | no polarisation, no Ring effect |
| | polarisation, no Ring effect |
| | no polarisation, Ring effect |
| vertical ozone profiles | China polluted |
| | South polar |
| | European background |
| | European background with uniformly polluted troposphere |
| | European background with with polluted boundary layer |
| Albedo | 0.1, 0.8 |
| solar zenith angle (at satellite) | 30°, 45°, 60°, 75°, 85° |
| viewing angle (at satellite) | 0°, 20°, 40°, 50°, 54° |
| realtive azimuth angle (at satellite) | 0°, 180° |

from Chance and Kurucz (2010) convolved with the measured instrument response functions. The signal to noise ratio is set in accordance with SNR values extracted from selected TROPOMI measurements. For each scenario, a TROPOMI measurement at possible closest conditions was selected to extract the SNR values. Depending on the wavelengths, viewing and illumination
geometry, typical SNR values for binned TROPOMI pixels vary between 100 and 600 in UV1 band and between 200 and 4000 in UV2 band. The noise sequences are created with a Gaussian random noise generator available within the SCIATRAN package. For each scenario, fifty independent noise spectra are generated. The synthetic retrievals use the same a-priori ozone profiles as for the real TROPOMI retrievals while the initial guess for the surface albedo is set to a fixed value of 0.5. Since neither calibration errors nor offsets are included into the simulations and the same RTM is used in the simulation and retrieval,
the calibration correction pseudo-absorber is turned off in the synthetic retrievals.

### 4.2 Sensitivity study

An overview of the ozone profiles resulted from the synthetic retrieval is shown in Fig. 2. The results for the five CAMELOT profile scenarios for all viewing geometry settings and both albedos are shown there. The three European profiles differ only at altitudes below 20 km and are intended to assess the sensitivity of the retrieval to the tropospheric pollution. In the stratosphere,
the ozone profile retrieval can reproduce true values very well. For the Southern Polar (D) and Chinese profiles (E), the a-priori profiles deviate from the true values by more than -25% or up to -1e18 molec/m³. Despite the large differences between a-priori and true profiles, the retrieval results are in a good agreement with the true profiles. Consequently, the retrieval seems to be nearly independent by the a-priori. Above 50 km, the sensitivity of the retrieval decreases and the deviations increase. This is explained by the fact that the maximum of the ozone weighting functions at the shortest wavelength lies at about 50 km. Below
25 km, the retrieval results strongly depend on the ozone profile scenario used. In general, the closer the a-priori profile is to





the true profile, the smaller the retrieval error is. In scenarios with small differences between the a-priori and true profiles in the troposphere, the retrieval results seem reasonable. Between 20 km and 25 km, all profiles show small negative deviations. Here, the vertical gradient of the ozone profiles is particularly strong and the retrieval is challenging because the averaging kernels have strong contributions from the ozone peak located above. Below 10 km, the deviations are within 25% for the

European profiles, (K) to (M), while for the other two profiles, (N) and (O), the a-priori seems to be too far from the truth, so that the retrieval results cannot reach the true values. For the same reason somewhat larger disagreement between the retrieved and true profiles is observed in the lowermost stratosphere (18 km to 25 km).

When the retrieval results are compared to observations or data products having a higher vertical resolution, the latter are usually convolved with the averaging kernels of the former. This also applies to the CAMELOT profiles used here as they have

a finer vertical structure than it can be resolved by the TOPAS algorithm. The convolution with the averaging kernels is done as follows:

$$\hat{x} = x_a + X_a \widetilde{A} X_a^{-1}(x - x_a) \tag{9}$$

where $x$ and $\hat{x}$ represent the original and the convolved CAMELOT profiles, respectively, $x_a$ is the a-priori ozone profile, and $\widetilde{A}$ is the averaging kernel matrix corresponding to the retrieval with the transformed variables (see Eq. (6)). In Figure 2 $\hat{x}$ is shown

in green. By applying the averaging kernels the comparison profiles are smoothed with the strongest changes occurring in the troposphere, where the vertical resolution of the retrieval (as shown in Figure 3) is lower. The difference between the retrieval results and averaging-kernel smoothed comparison profiles is significantly smaller than that without applying the averaging kernels. Overall, the deviations are below $\pm$ 10% over the entire altitude range.

As a next step, we analyse essential retrieval diagnostics, i.e. averaging kernel matrix (AK), vertical resolution, and precision.

They are shown for the European background profile (solar zenith angle (SZA) 30°, viewing angle (VA) 20°, relative azimuth angle (RAZ) 0°, albedo: 0.1) in Fig. 3. The averaging kernel matrix consists of 60×60 entries, one for each altitude layer. For the sake of clarity, AKs at every 5 km layers are shown in the panel (C). At altitudes between 20 and 45 km, the peaks are clearly pronounced and centred at the nominal altitudes of AKs. Above and below this region, the AK shapes are strongly asymmetric. For example, at 15 km there is no clear maximum. At this altitude, the retrieved profile is affected by changes

of the true profile in a wide range of altitudes with the strongest contribution from the 12 – 21 km altitude region and non-negligible influence from altitudes below 25 km to the ground. For the layers below 20 km, a double peak shape of AKs is observed. The averaging kernels for 0 km and 5 km do not differ much, which means that no vertical structure can be resolved in the lower troposphere.

The vertical resolution of the retrieval calculated as the inverse of the main diagonal of the AK matrix (Purser and Huang,

1993) is shown in panel (D) of Fig. 3. The best vertical resolution of about 6 km is reached in the stratosphere between 30 – 40 km. Above 50 km, the vertical resolution strongly degrades. As a consequence, we perform the TROPOMI retrieval only up to 60 km. In the upper troposphere at about 9 km, the vertical resolution locally degrades to about 20 km and then improves again with the decreasing altitude down to the boundary layer.



**Figure 2.** Retrieval results for the five CAMELOT scenarios (true), all settings are listed in table 2. The upper row, panels (A) to (E), shows the retrieved (orange), a-priori (black) and true profiles (red) as well as the true profiles convolved with the averaging kernels (green). Middle and bottom rows present the absolute (panels (F) to (J)) and percentage (panels (K) to (O)) difference between the retrieval results and the true profiles (orange), between the retrieval results and true profiles convolved with the averaging kernels (green), and between the a-priori and true profiles.



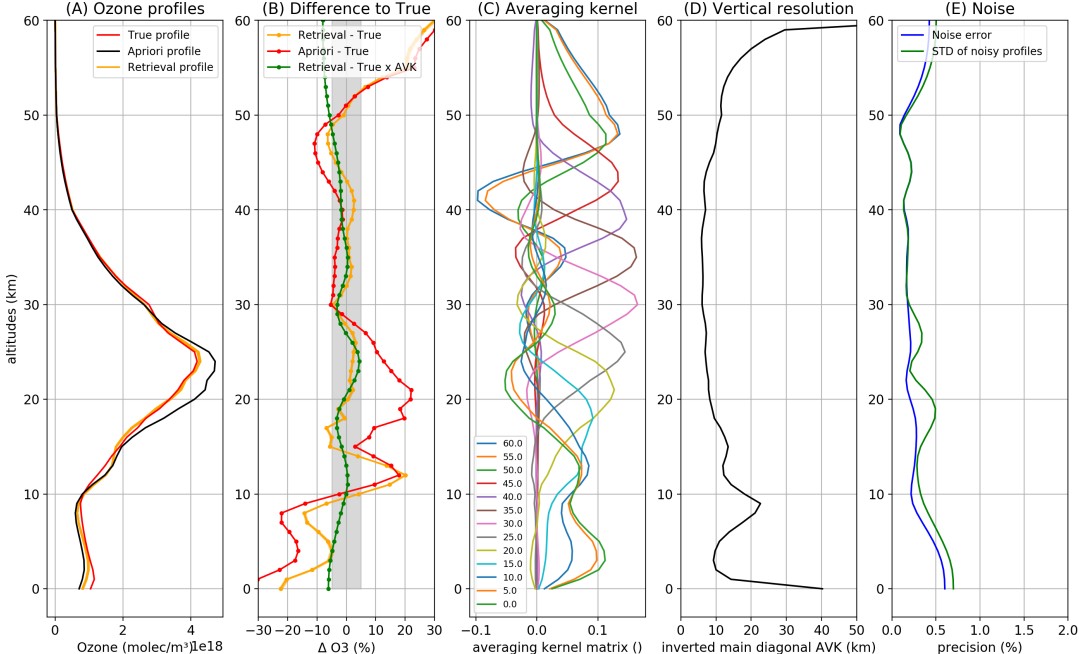

**Figure 3.** Retrieval diagnostics for the European background profile, SZA of 30°, VA of 0°, RAZ of 0°, and albedo of 0.1. Panels (A) and (B): Profiles and relative mean differences for 50 noisy spectra. Panel (C): Averaging kernels for every 5 km altitude levels. Panel (D): Vertical resolution of the retrieval. Panel (E): Retrieval noise error (blue) and total retrieval error (black) as defined by Rodgers (2002) as well as the standard deviation of the retrieval results from simulated spectra with 50 noise realisations (green).

The rightmost panel (E) of Fig. 3 illustrates the influence of the measurement noise on the retrieval results. The noise retrieval

error calculated in the linear approximation by using the Rodgers formalism is plotted in blue and is around 0.2%. The error is relatively small because the SNR values from binned TROPOMI pixels are used. In green, the standard deviation of the ozone profiles retrieved from synthetic spectra with different noise sequences added is plotted. It agrees well with the noise retrieval error. Following von Clarmann et al. (2020), we interpret our results as an estimate of the smoothed truth and thus do not consider the smoothing error as an error component.

Figure 4 shows the vertical resolution of the retrieval for all geometries and albedos for one true profile (polluted Chinese case). Each of the four panels shows the results for 25 combinations of the viewing angle (VA) and solar zenith angle (SZA) used in the sensitivity study. On the left, the retrieval results for low surface albedo (0.1) are shown, which simulates cloud and ice-free pixels. On the right, the high albedo (0.8) scenarios are shown. The top and bottom panels show the vertical resolution of the retrieval for azimuth angles of 0° and 180°, respectively. Between 18 km and 50 km, the vertical resolution is similar

for all geometries and surface albedos and ranges from 5 to 10 km. In the troposphere, the vertical resolution degrades to about 15 km (between 1 and 18 km altitude) but improves again in the lowermost layer (boundary layer). With exception of the uppermost and lowermost altitudes, the worst retrieval resolution of about 20 km is found at 10 km altitude. Differences between the vertical resolutions for the ground scenes with low and high albedo are most pronounced in the lower troposphere.

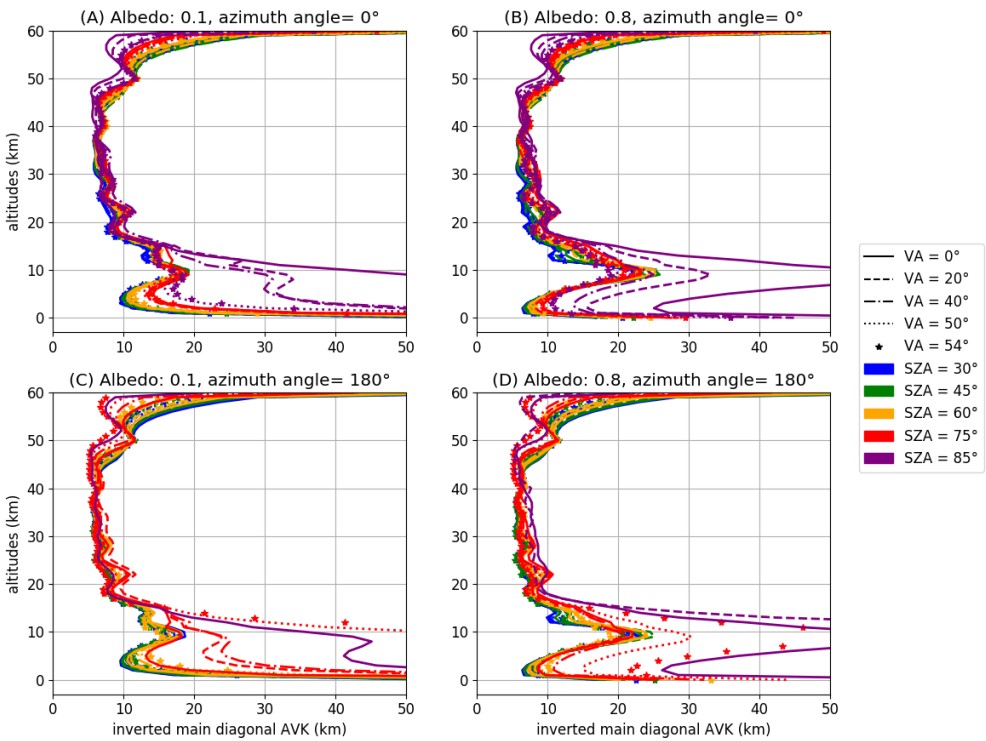

**Figure 4.** Vertical resolution of the retrieval for all simulations with the Chinese polluted ozone profile. The panels show the different combinations of the surface albedo (0.1 and 0.8) and relative azimuth angles ($0°$ and $180°$). Each panel displays the results for 25 combinations of VA und SZA.

The best vertical resolution in this altitude range is achieved for high surface albedo and small SZA. It should be noted, however, that a high albedo in the UV spectral range occurs only above clouds or snowy/icy surfaces. Ozone hidden below clouds can of course not be retrieved. The solar zenith angle (different colours) strongly affects the vertical resolution below about 17 km. The best vertical resolution is obtained at the smallest angles (blue). The vertical resolution is also impacted by the viewing angles (plotted with different line styles). At large SZA, there is an degrading vertical resolution with increasing VA below about 17 km. The troposphere becomes invisible for the retrieval at large SZAs ($> 85°$). Depending on the viewing angle, this might be the case already at $75°$ SZA for large azimuth angles.

A measure for the mean vertical resolution of the retrieved ozone profiles is provided by the number of degrees of freedom (DOF) determined from the trace of the averaging kernel matrix ($DOF = tr(\widetilde{A}) = \sum_{i=1}^{n} a_{ii}$). About 6.5 independent variables (from 6.3 for the south polar scenario to 6.7 for the polluted China case) can be retrieved in the altitude range between 0 and 60 km. This corresponds to a mean vertical resolution of about 9 km. Between 0 and 18 km the synthetic retrievals





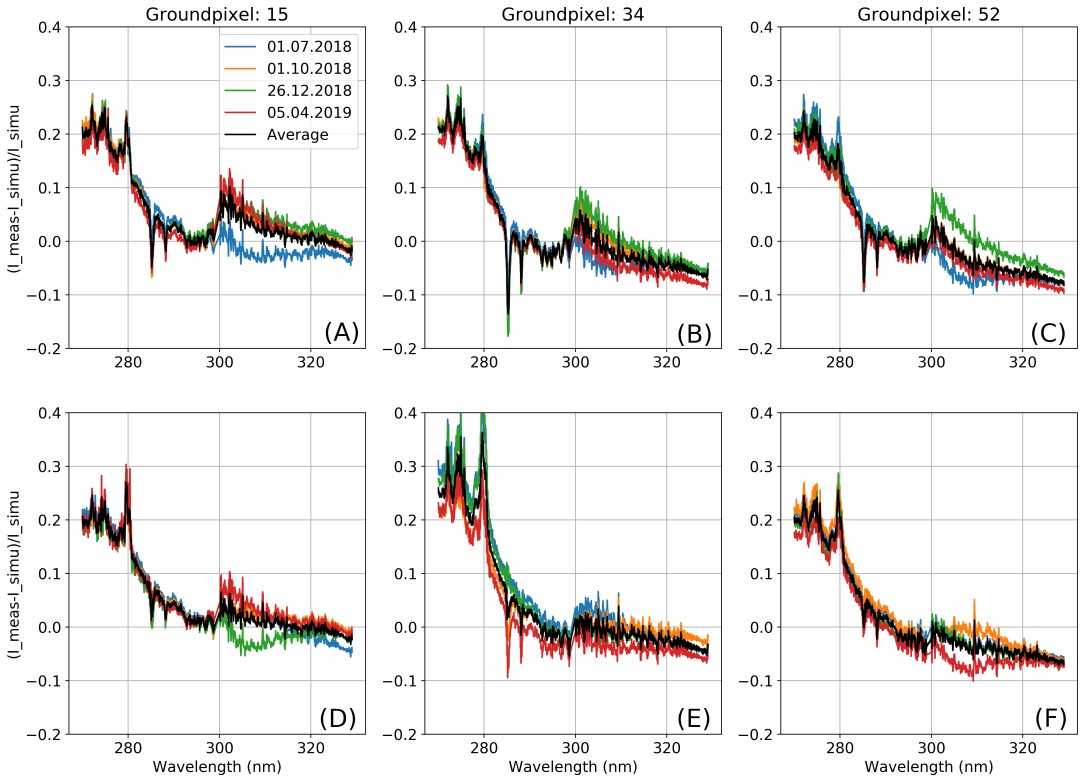

**Figure 5.** Relative differences between the measured and simulated radiances for three detector pixels (viewing angles). Upper panels: cloud-free tropical pixels ($-20°$ to $20°$ latitude). Lower panels: cloud-free extratropical pixels ($-50°$ to $-30°$ and $30°$ to $50°$ latitude). Black curves represent mean values of four orbits.

show around 1.5 DOF. This means that somewhat more than the total ozone content can be determined in the troposphere. In summary, profile information in the troposphere is severely limited and thus dependent on the a priori ozone profile.

It should be noted here that the number of independent variables might change in the retrievals using real TROPOMI data because of additional pseudo-absorbers which cannot be applied to the synthetic data. These pseudo-absorbers serve to compensate for inestimable calibration errors in the measured radiation.

## 5 Calibration correction for TROPOMI measurements

The UV channels of TROPOMI have been subject of intensive analysis and calibration before launch and during space operation (Ludewig et al., 2020). First tests with our retrieval applied to the most recent V2 TROPOMI spectral data revealed, however, large deviations in the retrieved ozone profiles, especially in the altitude range above 40 km. Therefore we decided to introduce additional spectral corrections as part of the retrieval. This type of corrections, often referred to as vicarious calibration or soft calibration, is described, for example, in Liu et al. (2010). The approach is to simulate TROPOMI radiance for a





selected set of measurements using ozone profiles from a comparison instrument, here MLS/Aura, and use the difference of the simulated to the TROPOMI measured radiance as the calibration correction. To this end, four orbits were selected at intervals of three months: orbit 3704 (1 July 2018), 5005 (1 October 2018), 6225 (26 December 2018) and 7642 (5 April 2019). The orbits were required to be cloud-free over the largest area possible. The spread of orbit dates allows us to check the stability of

the calibration correction with time. Forward simulations were performed with the SCIATRAN model for all cloud-free pixels (cloud content $< 10\%$). In contrast to the retrieval runs, the RTM calculations were performed with rotational Raman scattering and polarisation to make them as similar as possible to the measurements. The ozone profiles from the nearest MLS measurements were used to initialise SCIATRAN. Below the tropopause, the profiles were supplemented by MERRA-2 reanalyses (Gelaro et al., 2017) as already provided in the MLS data files. Allowing a maximum 1000 km between MLS and TROPOMI

pixels, we assume not to introduce a significant systematic error in the tropics, as ozone distribution is relatively homogeneous in this region. At higher latitudes, atmospheric dynamics may lead to higher variability. Here, TROPOMI measurements were only simulated if total ozone from MLS/MERRA-2 and TROPOMI (WFDOAS total ozone) agree to within 5%. The surface albedo (retrieved at 378 nm) was taken from the TROPOMI WFDOAS L2 product.

    The relative difference between the measured and simulated radiances is shown in Fig. 5 exemplarily for ground pixels

(viewing angles) 15, 34 and 52 (across-track ground pixel numbers of UV2). Since we identified a slight but stable latitude dependence in all radiance differences, we decided to use independent calibration corrections for the tropics and extra-tropics. In Fig. 5, the upper panels show the differences between the measured and simulated radiances for the tropical pixels ($-20°$ to $-20°$ latitude) while the lower panels illustrate the results for the extra-tropical pixels ($30°$ to $50°$ latitude in both hemispheres). The results from the four orbits are displayed in different colours. Overall, the relative difference in UV1 is significantly larger

than in UV2. However, the absolute intensities in UV1 are smaller by orders of magnitude. In the spectral range 270-280 nm the relative difference is about $+18$ to $+25\%$ reaching $+40\%$ in the middle of the swath (ground pixel 34) in the extra-tropics (panel (E)). The variation of the differences between the individual orbits is very small in UV1. In UV2, there are larger differences between individual orbits, as the sensitivity to albedo increases here. The variations between the orbits are the largest in the lower part of UV2 (shorter wavelength). This is because the sensitivity of the radiance to ozone decreases with the wavelength

while that to the albedo increases. As a result the maximum of the combined sensitivity is reached around 310 nm. Furthermore the surface albedo retrieved at 378 nm, used as input, may differ from the true albedo at shorter wavelengths. The difference is positive for all orbits at 300 nm (0 to $+10\%$) and tends to negative values (0 to $-8\%$) towards 330 nm. A closer look in the UV1 band shows a slight decrease of the differences with the time in all panels except for ground pixel 52 in the extra-tropics, panel (F). To account for this issue we calculate the calibration correction for each ground pixel and for the tropics and extra-

tropics independently as the mean relative difference spectra from all four orbits (black curves). The appropriate calibration correction is used then as a pseudo-absorber in the pre-processing fit procedure, i.e. a scaling of the calibration correction is allowed. The correction spectra derived from the tropical pixels ($-20°$ to $-20°$ latitude) are taken for that region exclusively, while the spectra derived from the extra-tropical pixels ($30°$ to $50°$ latitude) are applied to all pixels above $30°$ latitude in both hemispheres. Between $20°$ and $30°$ latitudes the correction spectra are interpolated.

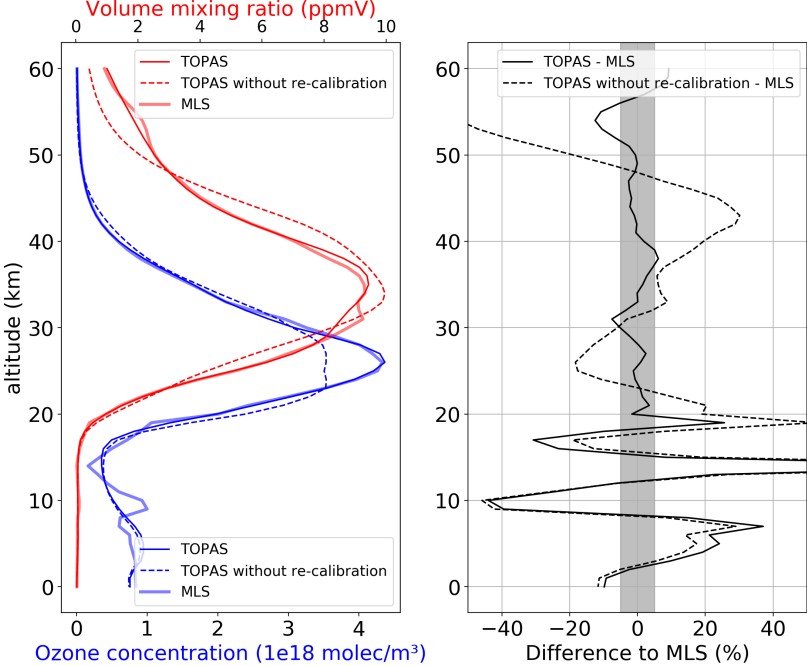

**Figure 6.** Comparison of ozone profiles from the TOPAS retrieval with MLS data for a single TROPOMI pixel (6 April 2019, orbit nr. 7664, $16.15°$ latitude, $-23.53°$ longitude, SZA: $20.15°$, VA: $11.47°$ and RAZ $6.60°$). Results for the TOPAS retrievals with and without calibration correction are shown. Left: Ozone concentration (blue) and volume mixing ratio (red) from TOPAS/TROPOMI and the collocated MLS measurement. Right: Relative differences between TOPAS retrieval and MLS. The relative differences are identical for both ozone concentration (blue) and volume mixing ratio (red).

Figure 6 demonstrates the effect of the calibration correction in the TOPAS retrieval. For a single TROPOMI pixel the TOPAS retrieval was carried out with and without our calibration correction. In this case a collocated MLS profile was available within 22 minutes and 28 km of the TROPOMI observation. The right panel of Fig. 6 illustrates the large difference between the ozone profile from the TOPAS algorithm without the calibration correction and the MLS results. The difference increases at high altitudes, which are more sensitive to short wavelengths (< 300 nm). The ozone profile from TOPAS including calibration correction agree very well with MLS between 20 and 50 km. The relative differences are within $\pm 5\%$.

## 6 Validation

To assess the accuracy and quality of the TOPAS algorithm, the retrieved ozone profiles were compared with ozone sondes and ozone lidar data. Both reference data sets have very good quality and a higher vertical resolution than ozone profiles from nadir satellite measurements (see Section 2). Ozone sondes are mainly used for comparisons in the troposphere and the lower stratosphere up to a maximum of 35 km. The middle stratosphere (15 - 50 km) is validated with lidar measurements.



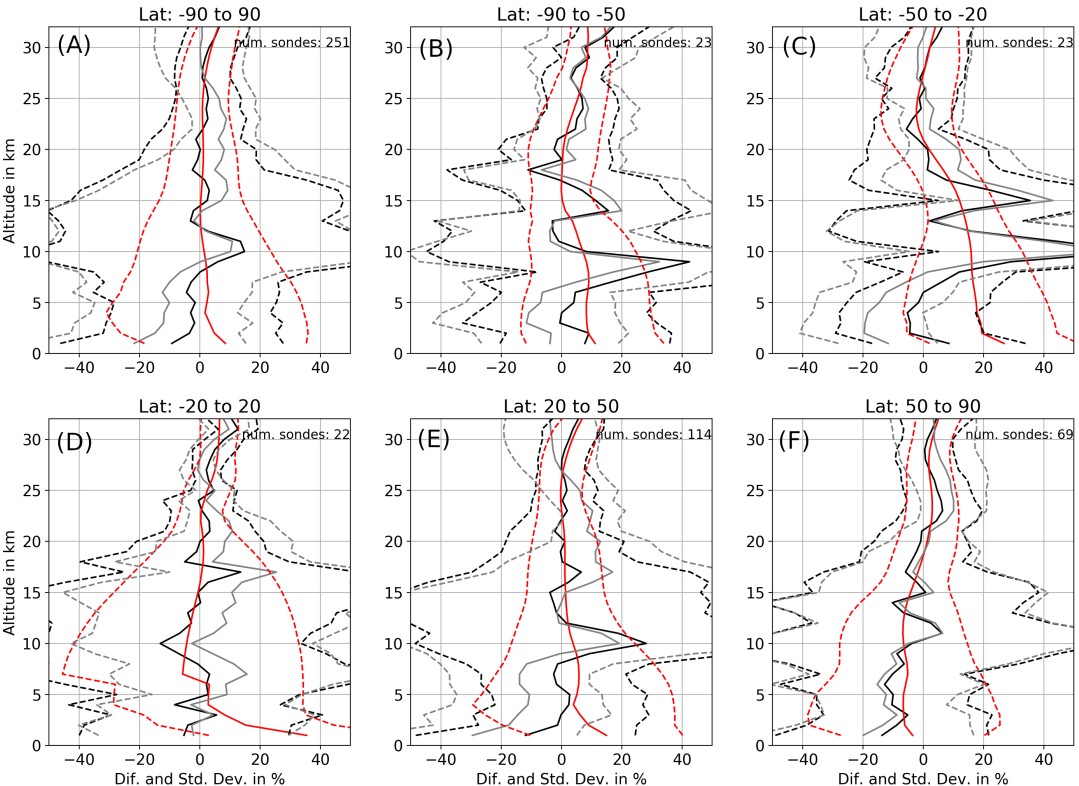

**Figure 7.** Comparison of the ozone profiles retrieved from TROPOMI and those from ozone sondes for different zonal bands. The relative mean difference between the retrieval results and the high resolution sonde data (solid line) as well as the standard deviation of the differences (dashed line) are shown in black. The comparison with the sonde profiles convolved with the TOPAS averaging kernels is shown in red. In grey the relative difference between the a-priori ozone profiles and high resolution ozone sonde profiles are displayed along with the corresponding standard deviations.

## 6.1 Ozone sondes

Figure 7 shows the results of the comparison of TROPOMI ozone profiles with ozone sonde measurements. Here, the relative mean difference $\Delta$ and its standard deviation $std(\Delta)$ is shown, which is defined by:

$$\Delta = \frac{\sum_{i=1}^{n}(r_i - s_i)}{\sum_{i=1}^{n} s_i}, \qquad std(\Delta) = \frac{n}{\sqrt{n-1}} \frac{\sqrt{\sum_{i=1}^{n}(r_i - s_i - \frac{1}{n}\sum_{i=1}^{n}(r_i - s_i))^2}}{\sum_{i=1}^{n} s_i} \qquad (10)$$

where $r_i$ and $s_i$ are the data from the $i$-th collocated pair of TROPOMI and sonde measurements, respectively. The relative mean deviation between the TOPAS retrieval results and high resolution ozone sonde data from all collocations (panel A, black curves) is about 5% down to about 13 km increasing to about 10% below. The standard deviation varies between 10 and 20% in the lower stratosphere (18 − 30 km) and between 25% and 50% below 18 km. The comparison of the retrieved O3 profiles from TROPOMI, convolved with the TOPAS averaging kernels (Panel A, red curves), and those from ozone sondes shows that



the relative difference is below ± 5% for all altitudes. The differences vanish in the altitude range where the retrieval is less sensitive.

The results for the northern latitudes and tropics (panels (D) to (F)), where the number of available sonde measurements are the highest, look very similar to one another. Above 18 km and below 8 km the relative mean differences are below about ± 10% with 15% standard deviation. At high and middle southern latitudes (panels (B) and (C)), the results are somewhat poorer in the

troposphere and lower stratosphere. Between 8 and 18 km differences of +40% and above can occur. Also noticeable are the larger differences between the a-priori and sonde profiles. As shown in Fig. 2, the results in the altitude range between 8 km and 18 km are strongly dependent on the a-priori ozone profile. A better choice of the a-priori in southern latitudes could possibly improve the retrieval results. With increasing latitudes the viewing geometries change and especially the SZA becomes larger. There is a connection between SZA and vertical resolution so that at larger angles the vertical resolution between 8 km and

18 km strongly decreases. As a consequence, the retrieval remains close to the a-priori.

In the lower troposphere (0 km to 5 km) an agreement within 10% is observed in all latitude bands, which coincides with the increased vertical resolution found in this altitude range, see Fig. 4. As the difference of the the a-priori to the sonde profiles (grey) is generally larger than that for the retrieved profiles, we can conclude that the retrieval improves the ozone knowledge in this altitude range for all latitude bands even if the standard deviation of the differences in the troposphere remains large.

Figure 8 illustrates the quality of the retrieval at different altitudes in a scatter plot. Again, the retrieval is compared with ozone sondes with (red) and without the convolution with the TOPAS averaging kernels (black). The linear regression is plotted as a solid line (red and black) and the dashed line marks the ideal 1:1 curve. At 1 and 5 km altitude the scatter is wide and also the R-value is small. At 10 km the convolution of the sonde profiles with TOPAS AKs improves the comparison, because here the information content is the lowest. The best agreement is found at altitudes 15, 20, and 30 km. The R-value here is above

0.8 (0.9 after convolving with AKs) and the linear regression is close to the 1:1 line. At 25 and 33 km the scatter is a little bit larger, but with R-values larger than 0.7 the agreement is still good.

## 6.2 Lidar

Figure 9 shows the relative mean differences (solid line) and the standard deviations (dashed line) between the retrieved profiles form TROPOMI and lidar data (black) for five individual stations and the all-station average (panel A). In addition, the

difference between the TOPAS results and the collocated MLS ozone profiles (convolved with TOPAS averaging kernels) is shown in green. When all stations are considered, the deviations between 15 and 45 km are below ± 5% (grey bar). There is a small positive bias between 25 km and 45 km that comes mainly from the Table Mountain station data set, which contains the largest number of comparison profiles. The standard deviation of the differences is about 10% in the range of 20 km to 40 km. The comparison with lidar profiles convolved with TOPAS averaging kernels (red) shows nearly the same results for the mean

relative difference, but a smaller standard deviation above 40 km and below 20 km. A closer look at the individual station reveals generally good agreement, but also individual station-dependent deviations. For OHP, Hohenpeißenberg and Lauder, no larger differences are identified. At Table Mountain there is a stronger negative peak of -9% around 20 km. Above this altitude a positive bias of up to +10% increasing with the height is observed. The comparison to MLS also shows the negative

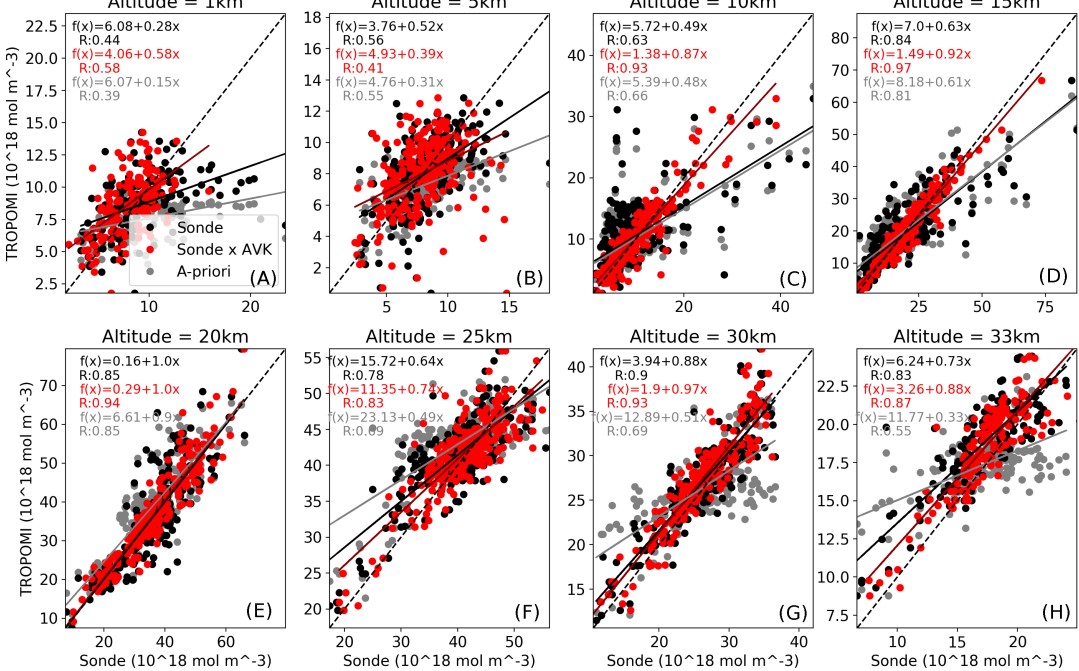

**Figure 8.** Scatter plot of collocated TROPOMI and ozone sonde profiles, the latter with and without convolving with TOPAS averaging kernels (red and black, respectively) for different altitudes. The comparison between a-priori and sonde profiles is shown in grey. The solid lines show the linear fit. The R-value is indicated in the top left corner of each panel (also black and red). The ideal 1:1 curve is plotted as a dashed line.

peak at 20 km but do not show the positive bias at higher altitudes. At Mauna Loa there is a positiv bias of +10% between 25
and 30 km that is also present in the comparison with MLS.

Scatter plots at individual altitude layers are presented in Fig. 10. The colours and lines are assigned in exactly the same way as for the ozone sonde validation shown in Fig. 8. From the scatter plots (Fig. 10) it is evident that in the altitude range between 15 and 40 km the agreement is very good with R-values above 0.6 both with and without convolving with TOPAS averaging kernels. An exception is the altitude of 25 km where the spread gets larger and the R-value lower. At altitudes above 45 km the
sensitivity of the TOPAS retrieval decreases and the agreement after convolving the lidar data with TOPAS averaging kernels is much better. It is notable that above 40 km the TOPAS retrieval shows smaller variability than lidar data. It is not yet known if this is a consequence of the lower sensitivity of the TOPAS retrieval or the lower precision of lidar measurements.

## 7 First results

As a first application, we show in Fig. 11 an example of a global distribution of TROPOMI ozone for 1 October 2018. Here, the
vertical structure of ozone is represented by the subcolumns within the following atmospheric layers: 0 – 8 km, 8 – 18 km, 18



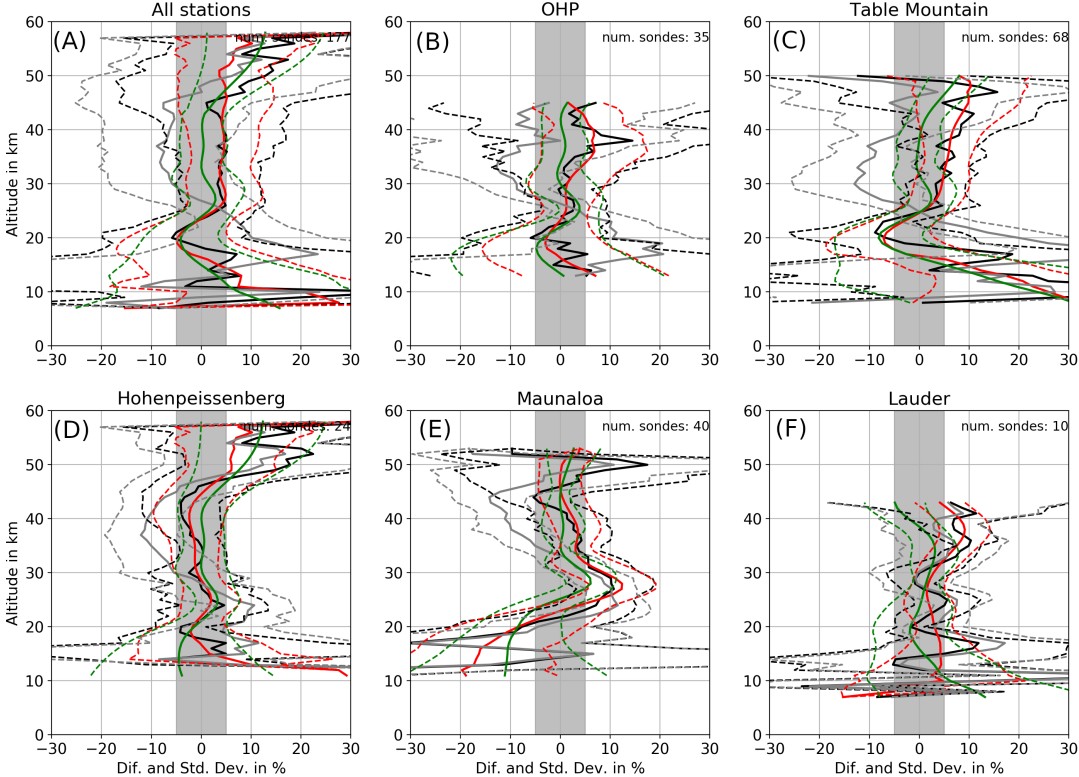

**Figure 9.** Comparison between TROPOMI ozone profile retrieval and the results from 5 different lidar stations and all-station average. The colors are same as in Fig. 7, but for lidar instead of ozone sonde data. In addition, the difference between the TROPOMI ozone profile retrieval and MLS results convolved with the TOPAS averaging kernels are shown in green. Only collocated MLS profiles are used (distance < 1000 km and time < 2h)

– 25 km as well as five layers with 5 km thickness between 25 and 50 km. The thickness of each layer roughly represents the vertical resolution of the retrieval in this altitude region in accordance with the results of the sensitivity study and validation. In the 0 – 8 km subcolumn, large variations of ozone are observed, which are believed to be a reason for higher standard deviations seen in the ozone sonde validation. These variations might be partially a retrieval artefact resulting from an insufficient retrieval

sensitivity in this altitude region but also reflect a natural high variability of the troposphere due to dynamic processes and local pollution sources. Furthermore, it is noticeable that areas with low ozone levels are related to the cloud coverage. In the 8 – 18 km layer, strong latitude gradients and rather large variabilities are observed. Most noticeable are large areas with high ozone density at middle and high latitudes, which can be identified as ozone streamers in the lower stratosphere and are related to dynamics in the atmosphere. In this altitude layer, the TOPAS retrieval has its lowest sensitivity in general, but a distinction

must be made between the lower and higher latitudes (see Fig. 12 (D)). While there is nearly no information content in the tropics, the ozone retrieval results in the higher latitudes are reliable. Between 18 and 25 km, the peak of the ozone number density profile is located and the range of ozone concentration variations is the largest. The largest ozone densities occur in the





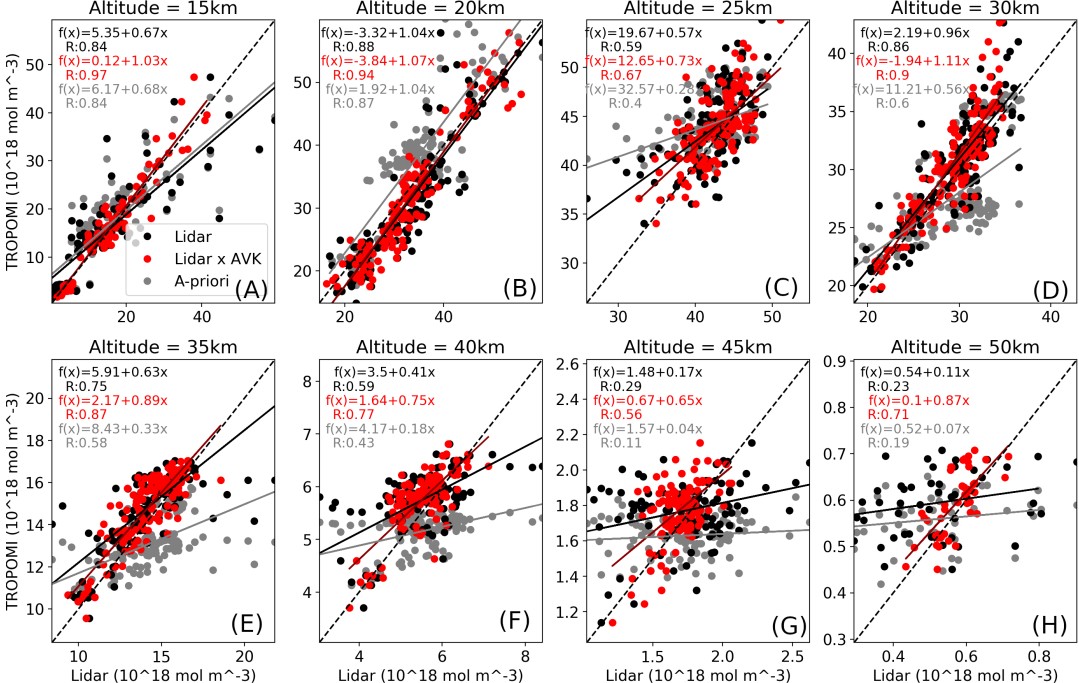

**Figure 10.** Same as Fig. 8 but for comparison of TROPOMI ozone profiles with lidar measurements.

southern mid-latitudes while the lowest concentrations are seen in the polar vortex. This is a typical situation in the southern hemisphere spring time. The ozone hole seen at polar latitudes is close to its largest extent at this time of the year. We note

that the complete coverage of Antarctica is not possible as TROPOMI does not measure during polar night. From 25-30 km layer upward, the latitude gradient of the ozone number density is reversed with maximum values occurring now in the tropics. At higher latitudes of both hemispheres rather low ozone values are observed. The layers 30 – 35 km and 35 – 40 km show similar ozone distributions as the layer below but appear much smoother and without jumps. That is to be expected because the retrieval sensitivity reaches its maximum at these altitudes. Also, according to the validation results, no significant systematic

errors are to be expected in these layers. The uppermost columns at 40 – 45 km and 45 – 50 km show the lowest overall ozone levels with a few small-scale variations, but a pronounced ozone maximum above the Southern Indian Ocean which extends into the southern Atlantic ocean.

From 25 km upward the south Atlantic anomaly (SAA) impact can be observed that results in larger scattering in the retrieved ozone values. The lower wavelength range (below 300 nm) is more strongly affected from the SAA which leads to

largest uncertainties in high altitude ozone. If necessary, the SAA region can be filtered out easily by the given flags in the TROPOMI L1B data product.

Panel (A) of Fig. 12 shows the zonal mean ozone number density as a function of the altitude and latitude for 1 October 2018. All fourteen orbits were binned at 1° latitude and averaged. The ozone maxima at about 25 km in the tropics and at about 20 km at southern mid-latitudes are clearly seen. Furthermore, a slight curvature of the ozone peak region with decreasing the peak



**Figure 11.** TROPOMI ozone subcolumns for selected atmsopheric layers for 1 October 2018. Vertical thicknesses of the layers roughly approximate the vertical resolution of the ozone profiles.

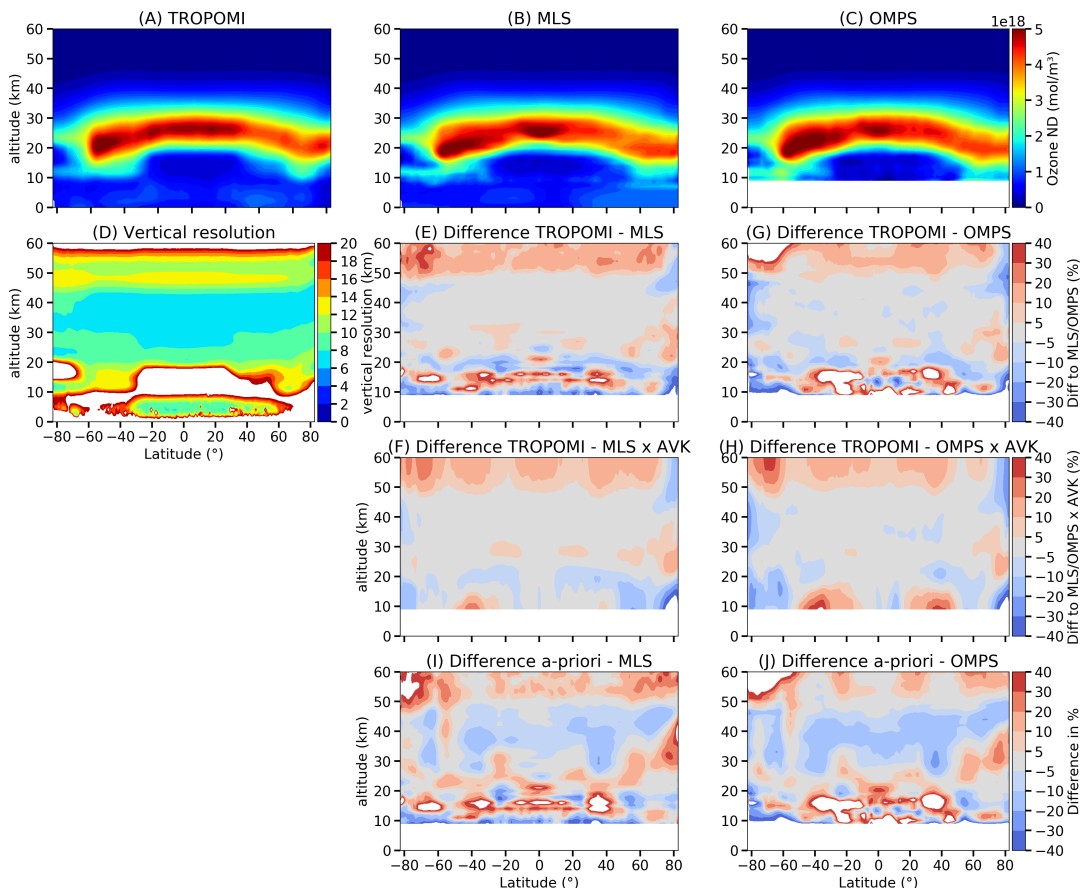

**Figure 12.** Panels A-C: the zonal mean ozone number density vertical distribution from TROPOMI (A), MLS (B), and OMPS (C) for 1 October 2018. Panel (D): the mean vertical resolution of the TROPOMI retrieval. Panel (E): the relative mean difference between TROPOMI and MLS results. Panel (F): the relative mean difference between TROPOMI and OMPS results. Panels (G) and (H): same as panels (E) and (F) but for MLS and OMPS data convolved with TOPAS averaging kernels. Panels (I) and (J): the relative mean difference between the TOPAS a-priori ozone profiles (climatology) and MLS (I) or OMPS (J) measurements.



altitude towards the poles is evident. In the high southern latitudes, the Arctic spring ozone hole is notable. The mean vertical resolution, shown in panel (D) of Fig. 12, is determined from the inverse main diagonal elements of the TOPAS averaging kernel matrix. Between 20 and 45 km, the vertical resolution ranges from 6 to 10 km. Below 20 km, it strongly depends on the latitude. Between $-50°$ and $50°$, the vertical resolution between 10 and 20 km is coarser than 20 km indicating that there is almost no information from these altitudes. Below 10 km it improves to 6-8 km between $-30°$ and $50°$ indicating that one

piece of independent information can be retrieved. The sensitivity study (Section 4.2) showed that the ozone profile retrieval below 18 km strongly depends on the used a-priori profile. Thus, in this altitude region, retrieved ozone profiles must always be considered with special caution. The middle and right panels of Fig. 12 show the zonal mean ozone number densities from the MLS and OMPS measurements, respectively, as well as their differences to the TROPOMI results. The closest MLS and OMPS profiles were selected for each retrieved TROPOMI pixel (maximum 1.5 hour time difference and 1000 km distance).

The direct comparison with MLS, i.e without convolving MLS data with TOPAS averaging kernels, (panel E) shows a good agreement between 20 and 50 km, with differences being mostly below $\pm10\%$. Between 20 and 30 km, there are several spots with the differences reaching $\pm20\%$. Above 50 km, where TOPAS retrieval has a low sensitivity, positive differences of up to +20% in the mid-latitudes and tropics are observed. At high southern latitudes, the positive differences increase to about 40% while for high northern latitudes, negative differences are seen. Below 20 km small scale patterns with higher differences

are evident. Below 15 km, the comparison results are less reliable as the precision of MLS data significantly decreases. The comparison with OMPS data (panel F) shows very similar patterns as for MLS. Overall the negative differences are somewhat more prominent. If MLS and OMPS ozone profiles are convolved with TOPAS averaging kernels to account for differences in the vertical resolution, the agreement significantly improves. For MLS data (panel G), differences above $\pm$ 10% only occur at high solar zenith angles (highest latitudes). Between 20 and 50 km, agreement below $\pm5\%$ is reached almost everywhere,

except for a narrow range around 30 km in the tropics and at high northern latitudes where positive deviations of up to +10% occur. Above 50 km, positive differences of up to +20% occur, while below 20 km mainly negative differences of up to -20% are observed. Again the comparison with OMPS data (panel H) is very similar to that with MLS, but the differences are somewhat larger. In the bottom panels, the comparison between TOPAS a-priori ozone profiles and retrieval results from MLS (panel I) and OMPS (panel J) is shown. From this, it can be seen that TOPAS retrieval improves the ozone profiles in all regions.

Finally, the retrieved albedo and the total ozone calculated from the retrieved vertical profiles are discussed. The total ozone column and albedo determined from TROPOMI WFDOAS are used to initialise the TROPOMI profile retrieval. The total ozone column information is used to select the proper a-priori climatological ozone profile. Figure 13 compares the retrieved albedo and the zonal mean total ozone from all retrieved profiles with the values from the TROPOMI WFDOAS retrieval as a function of latitude. The total ozone (left panel) agrees very well up to about 70°N. Both low total ozone values at high

southern latitudes and high values at middle latitudes are well represented. The mean difference is below $\pm2\%$ with a standard deviation of about 2%. Only at high northern latitudes larger differences of up to -9% are evident. As discussed above, at large solar zenith angles ozone in the lower atmosphere becomes less visible and thus not well retrieved. This then results in the contribution from the a-priori profile being larger in the total ozone column. The albedo (right panel) shows a stronger variation due to its natural characteristics. Again, very high values at high southern latitudes (due to ice) and low values in the





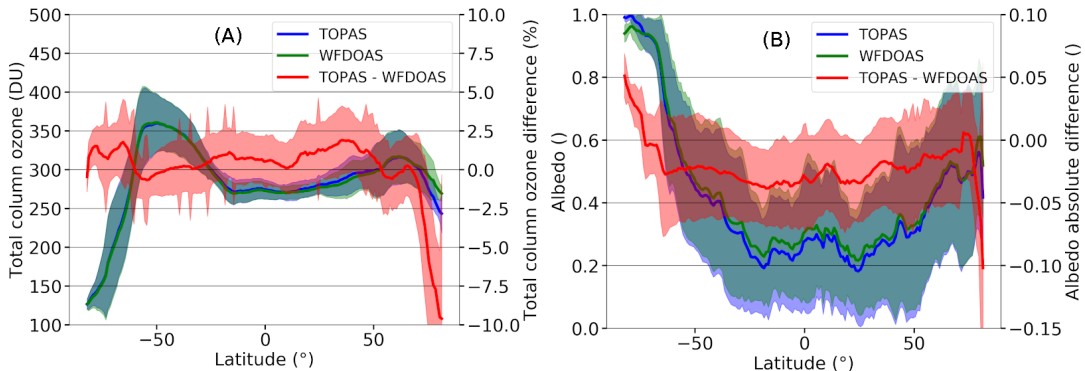

**Figure 13.** Comparison of the TROPOMI total ozone (left panel) and albedo (right panel) resulting from the TOPAS ozone profile retrieval and the WFDOAS total ozone product. The data were averaged over all orbits from 1 October 2018. The retrieval results are displayed in blue (TOPAS total ozone) and green (TROPOMI WFDOAS total ozone). Left y-axis is appropriate. Shaded areas mark the standard deviations of the means. The mean difference and the associated standard deviation are plotted in red (right y-axis is appropriate). Both albedo and total column from TROPOMI WFDOAS are used as a-priori assumptions in the TOPAS ozone profile retrieval.

tropics are well captured by both data sets. The mean difference shows a slightly negative bias of up to -0.025 with a standard deviation of about 0.025. However, it should be noted that the albedos in the different retrievals are not derived from the same wavelength ranges. Furthermore, disturbances in the ozone profile retrieval can influence or be compensated by changes in the albedo, since both are derived from the same wavelength range. Above 70°N (and high solar zenith angles), the albedo difference increases significantly up to $-0.1$.

## 8  Conclusion and summary


We developed a new TOPAS algorithm based on the first-order Tikhonov regularisation to retrieve the vertical distributions of ozone from TROPOMI measurements in the wavelength range from 270 to 329 nm. In the sensitivity study, we estimated the retrieval quality using synthetic spectra adapted to TROPOMI. We found that the quality of the ozone profile retrieval depends on the viewing geometry, albedo, and the a-priori ozone profile. In optimum cases (small solar zenith angles and large

albedo), deviations of the individually retrieved profiles from the truth are within $\pm\,5\%$ in the stratosphere. In the troposphere, the agreement is strongly dependent on the a-priori profile. Compared to the a-priori ozone profile, the retrieval improves the result, but can barely compensate differences of more than $\pm25\%$ between the a-priori and true profiles. If the true profile is convolved with the averaging kernels, differences are generally within $\pm10\%$ in the entire altitude range considered (0 – 60 km). For all simulated scenarios the vertical resolution of the retrieval is 6 – 10 km in the stratosphere (18 – 50 km) getting

worse in the lowermost stratosphere and the troposphere (up to 15 km). In the troposphere, the vertical resolution degrades with increasing solar zenith angle. Above 75° SZA, almost no information from the troposphere is available. The number of

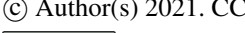



independent variables (degrees of freedom) which can be retrieved over the entire altitude range is around 6, which corresponds to a mean vertical resolution of 10 km from 0 to 60 km altitude.

In addition to improvements in the UV radiometric calibration in Version 2 of TROPOMI level 1 data (Ludewig et al., 2020), a radiometric calibration correction was determined by using radiative transfer simulations and MLS ozone profiles. For the shortest wavelengths, radiometric adjustments of +20% and more are needed. The correction drops below $\pm$ 10% for longer wavelengths.

   The TOPAS ozone profile retrieval was validated by comparisons with ozone sonde and stratospheric ozone lidar data. For the limited TROPOMI data set available so far, we found very good agreement with both validation datasets. The validation

with ozone sondes shows good agreement in the lower stratosphere with deviations of less than $\pm$10% at all latitudes in the altitude range 18-30 km. The standard deviation of the mean differences is about 10%. Below 18 km, the mean difference and the standard deviation increases strongly but decreases when the sondes are convolved with the TOPAS averaging kernels.

   The validation with lidar measurements shows a bias within $\pm$5% between 15-45 km. The standard deviation is about 10%. The convolution of the lidar profiles with TOPAS averaging kernels barely changes the result because of a high sensitivity and

relatively good vertical resolution in the stratosphere.

   By applying the TOPAS retrieval to a full day of TROPOMI measurements (14 orbits), we demonstrated the high potential of the ozone profile retrieval, enabling us to determine highly spatially resolved vertical profiles. Important dynamical features in the atmosphere such as streamers and seasonal events such as the ozone hole can be detected. A comparison with ozone profiles from MLS and OMPS limb measurements show typically good agreement to within $\pm$5% between 20 and 50 km. If

the averaging kernels are taken into account, the differences above 50 km and below 20 km are also getting small, reaching about $\pm$20%. Good agreement in total ozone compared to TROPOMI WFDOAS (within $\pm$ 2%) was found.

   When a continuous L1B version 2 data set is made available for TROPOMI, we plan to close the data gaps by processing this data. This will enable us to extend the validation using a larger data set from early 2018 until present and to investigate time series. The verification of the temporal stability of the ozone profile retrieval is still ongoing. The assessment of the calibration

correction using MLS data as implemented in this study is also of particular importance in this context. If the finding of this study, that a temporally independent but spectrally dependent radiometric correction factor is sufficient and holds for the full data set, the final data product can be created which is independent of MLS apart of the initial correction. That would enable the use of TROPOMI ozone profiles for trend analysis and monitoring. Thus, the goal to use TROPOMI measurements to mitigate consequences of a possible lack of limb instruments may become feasible.

*Data availability.* All data are available at the University of Bremen upon request.

*Acknowledgements.* This study was supported and funded by the BMWi/DLR project 'S5P Datennutzung' (Förderkennzeichen 50EE1811A), the University of Bremen, and the federal state Bremen. The S5P L1b version 2 data is provided by ESA/KNMI via the S5P validation team



(S5P-VT) activities. We acknowledge all ozone sonde providers and their funding agencies for performing regular sonde measurements and thank the WOUDC and SHADOZ network archiving these data. The same applies to the teams from all lidar stations we used. Part of
this research work was carried out at the Jet Propulsion Laboratory, California Institute of Technology, under a contract with the National Aeronautics and Space Administration (80NM0018D004). Ozone profiles of the MLS limb measurements are provided by NASA.



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
