# Peer review of "Ozone Profile Retrieval from nadir TROPOMI measurements in the UV range"

_Atmospheric Measurement Techniques, 2021_

## Author Comment (AC3)

**Answers to reviewer 2**

This paper presents a new TROPOMI ozone profile retrieval from UV (UV-1 and UV-2) measurements using the Tikhonov regularization based TOPAS algorithm. A sensitivity study is presented to show the retrieval quality from synthetic data. Systematic biases in TROPOMI data are shown by comparing simulation using MLS+MERRA-2 ozone profiles with observation, and soft calibration is derived and applied to the retrievals. The retrievals are validated with ozonesonde and lidar measurements as well as MLS, and OMPS satellite measurements. Examples of global distribution of ozone at different altitude ranges are shown. The scope of this study is well suited for AMT. This paper is generally well organized and methodology is generally good. However, some of the important retrieval details are not described and some places require clarification. Overall, I think that this paper can be published after addressing the specific comments below.

Specific comments:

1. **L38, SAGE-III on ISS was launched in 2017, not 2006. Suggest changing to "launched in 2001 on Meteor-3M and in 2017 on ISS"**

It is corrected in the text.

2. **L77, I think that it should be "in-flight analyses"**

Yes, Antje Ludewig is also writing about in-flight calibration. It is corrected in the text.

3. **L81-84, it would be useful to describe a little more about ozone profile retrievals by Zhao here in addition to the spectral range: for example, using the optimal estimation method, also derive and apply soft calibration. Band 3 is used due to larger systematic radiance differences in band 1 and larger fitting residuals in band 2, larger biases in total ozone and ozone profile with relative to other correlative measurements.**

We have extended the description of the ozone profile retrieval by Zhao.

*"So far, the L1B version 1 TROPOMI data in band 3 (314 - 340 nm) was used by Zhao et al.,2020 to determine tropospheric ozone and investigate its changing distribution due to the Covid-19 pandemy. Their profile retrieval was limited to the UV3 band because of larger systematic radiance differences in band UV1 and larger fitting residuals in band UV2. The ozone profiles were derived using Optimal Estimation and a soft calibration was applied as well. Due to the narrow spectral window the vertical resolution of their retrieval is very limited (1.5 - 2 degrees of freedom)." (Line 83-88)*

4. **L113, should the spatial resolution of retrievals at nadir be 28.8 x 45 km2 due to coadding of 8 UV2 across-the-track and 8 pixels along the track?**

For band 2 (UV2) an internal binning factor of 2 is already applied in the L1B product. These should result in a spatial resolution of around 45x45 km² (at nadir-viewing point).

(no correction in paper)

5. **L130, according to how SNR is calculated in the L1b (mostly Poisson noise), I think that the binned error should be 1/SQRT(n) * SUM(SNR_i). If n=1, according to your equation, SNR_binned = 1/SQRT(2) * SNR rather than SNR.**

This is correct. We use SNR_binned = sqrt(1/n * sum(SNR^2)). The formula is corrected in the text.

6. **L150, please check the reference as Flynn et al. (2014) does not talk about LP retrievals. A better reference to the LP measurements may be Jaross et al. (2014): https://agupubs.onlinelibrary.wiley.com/doi/full/10.1002/2013JD020482**

Yes this is true. We have changed the citation.

7. **L167, for the temporal collocation criterion, it says "24 hours time differences", as it is a fixed time difference here, suggest changing to "within 24 hours" or "24 hours maximum time differences". Also, I guess most of the time differences are within 12 hours. You may tighten this criterion.**

We added 'maximum' to the time difference to provide confusion.

And yes most of the sonde measurements are within 12 h (91%) and we could tighten the criterion, but we don't see a negative effect using them and the dataset is already very limited.

8. **L184, the full name of TOPAS has been mentioned earlier in the introduction, so you do not need to mention here again.**

It is changed in text.

9. **Table 1 is not referenced in the text. You may mention Table 1 at the beginning of Sect. 3.2 before describing the retrieval in more details.**

We added a sentence to reference the table.

10. **L215, you mad add "base on LUT" after "A polarization correction" and add " and will be described later in this section"**

We have included the additions.

11. **L216, based on the text below, offset is also fitted. So you add it to this sentence. Why is a 1$^{st}$ order polynomial subtracted? L256 also does not mention why and what kind of 1$^{st}$ order polynomial is subtracted. Is the 1$^{st}$ order polynomial pre-determined?**

We added the offset to the sentence in Line 223.

A first order polynomial between 300 nm and 310 nm is determined by the least squares fit and subtracted for each fitting term because this wavelength region is most critical for the retrieval. There is a kind of discontinuity around 300 nm shown in Fig 5. As the scaling of the recalibration term is not sufficient to remove the discontinuity near the channel boundary, a linear polynomial is subtracted in addition. The polynomial degree (1st order) was selected after tests using different polynomial orders.

*"This pseudo-absorber, as shown in Fig 5, indicates a discontinuity around 300 nm. In the second spectral window (300 - 310 nm), the scaling of the re-calibration term is not sufficient to remove the discontinuity near the channel boundary. Therefore, in this spectral window, a linear polynomial is fitted by least squares and subtracted for each fitting term separately."* (Line 266 - 269)

**12. In table 1, it is not clear about how and why Tikhonov 0$^{th}$ order parameter of 11.11 is set. According to equation (4), the 0$^{th}$ Tikhonov term is just Sa^-1.**

Exactly, the Tikhonov 0$^{th}$ term is just Sa^(-1) and with a given constant a-priori variance of 30 % it becomes 11.11 (1/0.3²). We just add that value for the sake of completeness to Table 1.

(no changes in text)

**12. L255, what do you mean "represented by the inverse solar spectrum"? Are you fitting a scaling factor to the solar spectrum? Please clarify it.**

We are using the inverse solar spectrum as a pseudo-absorber and fit it to the radiance measurements. For our three spectral windows we fit then three scaling factors.This corresponds to a wavelength independent offset correction for each of the spectral windows. Further details can be found in:
Rozanov, A., Weigel, K., Bovensmann, H., Dhomse, S., Eichmann, K.-U., Kivi, R., Rozanov, V., Vömel, H., Weber, M., and Burrows, J. P.:Retrieval of water vapor vertical distributions in the upper troposphere and the lower stratosphere from SCIAMACHY limb measurements, Atmospheric Measurement Techniques, 4, 933–954, https://doi.org/10.5194/amt-4-933-2011, 2011

*"The third pseudo-absober accounts for a wavelength independent offset in the radiance spectra. To this end, the inverse of the solar spectrum is fitted to the radiance spectra with one scaling factor for each of the three spectral windows (Rozanov, 2011b)."* (Line 269)

**14. L264, you may change to "in generally good agreement on altitude average" as the actual variance changes a lot.**

Added in the text.

**15. L268, the gamma value is fixed to 0.007 for all the retrievals here. In the Tikhonov method, the regularization parameter (i.e., gamma here) is often determined dynamically, for example, using the L-curve method. Have you tried to derive it dynamically?**

The Tikhonov regularisation term is carefully determined in order to be suitable for all retrieval conditions for TROPOMI measurements. Indeed, there are proven methods like L-curve to optimize single profiles, but we decided not to use a dynamical method to assure the stability of the retrieval.

(no changes in text)

**16. L272-273, has the noise in the solar irradiance been included in the calculation of SNR?**

No, we haven't included the solar irradiance noise. The different orders of magnitude of an example TROPOMI pixel are shown in Fig. 1 (of this document). The (binned) pixel is located at -51° lat, 145° lon, 56° SZA, 9° VA and 70° AA with an albedo of 0.1. In green the solar irradiance noise is shown. In comparison to the binned radiance SNR (blue) the irradiance SNR is 2 to 0.8 orders of magnitudes larger. Especially in UV1 the difference between the binned radiance SNR (around 100) and the irradiance SNR (around 10,000) is huge. Therefore we decided to neglect the irradiance SNR in our TROPOMI retrieval.

[Figure]

*Figure 1: SNRs for a  single (binned) TROPOMI pixel (orbit 5005) at 51° latitude, 154° longitude, 56° SZA, 9° VA and 70° AA with an albedo of 0.1. In green the irradiance SNR is given,  which is binned for 8 pixels in UV2. The binned (1x8 UV1, 8x8 UV2) radiance SNR is shown in blue and the mean radiance SNR is given in yellow. The (binned) SNR, which was used for the sensitivity study  with simulated spectra shown in Fig. 2 in the paper, is shown in black.*

17. **L277-278, it is not clear about how "no cross-talk between these two parameters is considered while both are retrieved in one iterative step. I guess that the covariance terms in Sa are 0 between ozone and surface albedo parameters. Do you also set the Jacobians for effective surface albedo to 0 below 310 nm?**

Yes, we meant that the covariance between ozone and surface albedo in Sa is 0 and the Jacobians for the effective surface albedo below 310 nm set to are zero.

18. **In Fig. 2, do the green curves on middle and bottom panels show the mean differences between retrievals and convolved true profiles? Or do they show the differences between the mean retrievals (orange with circles) and convolved true profiles? It is not clear from the figure caption.**

The mean differences between retrievals and convolved true profiles are shown. We have clarified the figure caption.

19. **In Fig. 3 caption, the total retrieval error (black) is not seen and also not shown on the legend of panel (e). Should the layer legends on Fig. 3c be 0.5, 5.5, … 59.5 km. From the text, the retrievals are done at 60 layers, but on Figs. 2 and 3, the results are plotted at 61 levels from 0 to 60 km. Please clarify this.**

We decided not to show the total retrieval error (and removed it from the caption) because it contains the smoothing error and should no longer be mentioned. (Thomas v. Clarman, 'Smoothing error pitfalls', 2014, https://doi.org/10.5194/amt-7-3023-2014 )

We have corrected the text and it now says '61 levels'. From our definition this also means 60 layers, but it is less confusing. Figures 2 and 3 are plotted correctly with 61 levels.

20. **L360, based on the definition of vertical resolution as the inverted main diagonal (i.e., layer degree of freedom for signal), the vertical resolution becomes smaller if the retrieval is done at a coarse grid (e.g., every 5 km). Does this definition require the retrieval to be    done at every 1 km or require the normalization to layer thickness?**

The definition of the vertical resolution given by the inverted main diagonal can also be used for finer/coarser grids. In this case, the inverted main diagonal elements of the averaging kernel matrix have to be multiplied by the vertical grid step.

21. **L365 and Fig. 3e, the noise errors and standard deviations of retrieval results seem to be too small to be true especially in the lower stratosphere and**

**troposphere. Please check them. What are the binned SNRs at different wavelengths for this specific spectrum?**

The binned SNR for the specific simulation is shown in Fig. 1 (this document). The simulated retrieval shown in Fig. 2 (30° SZA, 0° VA and 0° AA) is one of the best cases in terms of viewing geometry and SNR. A high SNR is expected and this should result in a very low noise error. Figure 1 shows a higher simulated SNR (black) in comparison to the single TROPOMI pixel (blue), but it is still in the same order of magnitude. We cannot find a reason why the low noise error should be too small.

22. **L375-376, it cannot be seen from Fig. 4 that the vertical resolution improves for the bottom layer as the values become much larger for the bottom layer**

The vertical resolution improves in the 5 km altitude region (and is the worst for the bottom layer). We have corrected that in the text.

23. **L383-385, based on Fig. 4, the sentence "At larger SZA, … with increasing VA below 17 km" is true for relative azimuthal angle of 180, but opposite for relative azimuthal angle of 0. Can you please explain why the vertical resolution depends so greatly on VA for large SZA and the dependence is on the opposite? The troposphere becomes invisible at SZA of 85 except for VZA 50/54 and relative azimuthal angle of 0, right? Why the vertical resolution significantly increase for VZ 50/54? Is this real or due to some kind of anomaly in the averaging kernels?**

The vertical resolution, which is shown in Figure 4, is determined by the ozone absorption in each particular altitude region on the total strength of the ozone absorption signal registered by the instrument. Two factors are essential here, both dependent on the observation/illumination geometry. These are the length of the effective light path through the altitude layers of interest and the amount of light originating from these layers. While the latter depends on the amount of the incident light and scattering properties of the atmosphere, the light paths always increase with the geometrical angles (solar zenith angle (SZA), viewing angle (VA) and azimuth angle (AA)). Furthermore, the simulations were done using TROPOMI specific SNR values for each viewing geometry, which also influences the sensitivity of the retrieval. As follows from Fig. 4, generally less light penetrates into the troposphere at large solar zenith angles reducing the vertical resolution. For large AA we found that the vertical resolution is better for smaller VA, which is most probably related to an increased light path of the direct solar light through the troposphere thus decreasing the amount of light to be scattered. The opposite effect takes place for smaller VA.

We added the explanation to the text:

*"In general, the vertical resolution is determined by the ozone absorption in each particular altitude region on the total strength of the ozone absorption signal registered by the instrument. Two factors are essential here and both depend on the observation/illumination geometry. These are the length of the effective light path through the altitude layers of interest and the amount of light originating from these layers. While the latter depends on the amount of the incident light and scattering properties of the atmosphere, the light paths always increase with the geometrical angles (SZA, VA and AA). Furthermore, the*

*simulations were done using TROPOMI specific SNR values for each viewing geometry, which also influences the sensitivity of the retrieval.*

*The solar zenith angle (different colours) strongly affects the vertical resolution below about 17 km as less light penetrates into the lower atmosphere at large solar zenith angles. The best vertical resolution is obtained at the smallest SZA (blue), where more light is scattered back to the instrument.*

*The vertical resolution is also impacted by the viewing angles (plotted with different line styles) and the azimuth angles. Below about 17 km and at large SZA, there is a degrading vertical resolution with increasing VA for very high AA, which is most probably related to an increased light path of the direct solar light through the troposphere thus decreasing the amount of light to be scattered.. On the contrary, for lower AA the vertical resolution is improving with larger VA."* (Line 400 - 412)

**24. L392-394, how and how much do the additional independent variables generally change the DOF?**

The DOFs are based on AKs, so they are not directly affected by the pseudo-absorbers because they are fitted in the pre-processing. However, there may be a change if the pseudo-absorbers strongly modify the ozone content and thus the vertical resolution.

**25. L466-499, the first sentence say "the increased vertical resolution found in this altitude range, see Fig. 4." But according to Fig. 4., the values at few bottom layers increase (compared to 3-5 km?) and thus the vertical resolution become worse. Also for some of the larger SZAs or high latitude, the vertical resolution decreases in this altitude range and the retrieval sensitivity is very limited in 0-5 km. So I think that this statement is not accurate. Also to show the retrieval improvement over the a priori, I think that it is equally important to show the improvement of standard deviation of the differences. So at what altitude/latitude ranges are the standard deviations of the differences between retrievals and ozonesonde better than those between a priori and ozonesonde?**

As mentioned in comment 22, the vertical resolution is improved around 5 km and not at the bottom layer. Furthermore we have pointed out that the improvement in the lower troposphere is not significant. The standard deviation shows nearly no change so that we don't see a gain in the standard deviation although we can observe a reduction of the bias.

*"As the difference of the a-priori to the sonde profiles (grey) is generally larger than that for the retrieved profiles, we can say that the retrieval improves the ozone knowledge in this altitude range for all latitude bands. But we have to note that the standard deviation, in comparison to the a priori, remains almost the same."* (Line 503 - 506)

**26. L470-476, it is also useful to discuss about the slope and whether the retrievals improve over the a priori.**

We added a short discussion about the regression line in the sonde and lidar scatter plot.

*"The slope of the linear regression lines (black and red) improves for each shown altitude in comparison to the a-priori slope (grey). In regions where the information content is very limited (10 and 15 km) only the comparison to ozone sondes with applied AKs show an improvement. " (Line 513 -515) and "Similar to the ozone sonde results, the slope of the linear regression is improving in comparison to the a-priori." (Line 537 - 538)*

**27. Figure 9 legend, "num. sondes" should be "num. Lidar"**

Fig. 9 is corrected.

**28. L496-497, it is likely due to reduce retrieval sensitivity for these altitudes? What are the precision for lidar measurements at these altitude ranges.**

A new study about the ozone lidar in Hohenpeißenberg by Wing et al. ( 'Evaluation of the new DWD ozone and temperature lidar during the Hohenpeißenberg Ozone Profiling Study (HOPS) and comparison of results with previous NDACC campaigns', 25. May 2021, https://doi.org/10.5194/amt-14-3773-2021) cites an uncertainty of around 5% up to 40 km. Above 40 km, the uncertainty is strongly increasing (Fig. 14).They found the same bias pattern (+20% to +40% above 45km) as we do, when they compare the ozone lidar profile to other satellite ozone profile products like MLS and SABER (Figures 6 and 14 in their publication). Therefore we assume that our retrieval provides reliable results in the altitude region above 40 km even if the sensitivity is reduced.

**29. L506, are you getting a priori ozone between scene pressure and surface pressure? Or do you retrieve ozone only above scene pressure?**

We only use one a-priori value for the lowest level and are retrieving ozone only for the lowest grid level.

**30. L503-506, it is worth to add that the 0-8 km sub-columns show high wave one pattern in the tropics, with high ozone in the South Atlantic and low ozone in the tropics, and generally higher ozone at mid-latitudes as these features are generally consistent with the tropospheric ozone distribution.**

We add the description of tropospheric ozone to the text.

*"In the 0 - 8 km subcolumn (A), large variations of ozone are observed, which are believed to be the reason for higher standard deviations seen in the ozone sonde validation. This also reflects the high natural variability in the troposphere due to dynamic processes and local pollution sources. In general the tropical wave one pattern is well reproduced (low ozone in the Pacific, higher ozone in the South Atlantic). Furthermore, it is noticeable that areas with low ozone levels are related to regions with high cloud coverage. " (Line 543 - 548)*

**31. L510 and L534, I suggest changing to "no vertical information" as there is still useful information for the sub-columns.**

We changed the passage to "no information of vertical structure".

**32. L562-563, this seems to suggest that the TOPAS total ozone at high latitudes is less accurate compared to WFDOAS total ozone. I think that WFDOAS**

**retrievals might be more sensitive to a priori ozone as no vertical     ozone information is retrieved.**

At high solar zenith angles both algorithms have increased uncertainties due to loss of sensitivity to the lower layers in the atmosphere. As WFDOAS uses longer wavelengths (>325 nm) than TOPAS (> 270 nm), the SNR loss with increasing solar zenith angle more strongly affects TOPAS and may result in higher uncertainties at highest SZAs.

**Technical comments: (all corrected)**

1. **Title: Some words (not prep. or adv.) are not capitalized: nadir, measurements, range. You may remove "nadir" as TROPOMI is a nadir-viewing only instrument.**

   We checked the AMT submission regulations and found that titles should follow sentence-style capitalization. Therefore, we now use lower case letters. We keep "nadir" because not all readers may immediately know that TROPOMI is a nadir-only instrument.

2. **In abstract, it might be useful to show the unabbreviated name of "TOPAS" algorithm at its first occurrence.**   corrected
3. **In abstract, you may show full name of "TROPOMI" as "Tropospheric Monitoring Instrument (TROPOMI)", "MLS" as "Microwave Limb Sounder (MLS) on the Aura satellite", and of "OMPS-LP" as "Ozone Mapping and Profiler Suite Limb Profiler (OMPS-LP)" at their first occurrences**     corrected
4. **L51, change to "higher" and use subscript for 3 in O3**   corrected
5. **In equation (6), S_R should be S_r for consistency.** corrected
6. **L220,   there is an extra "." after "a priori"**          corrected
7. **L225,   miss an "." Before "The effective"**          corrected
8. **L228, good to specify the   unabbreviated names of OCRA and ROCINN.** corrected
9. **L271,   "y" in "Sy" should be in subscript**          corrected
10. **Figure 2 caption, add 's' to "difference"**   corrected
11. **L328, change to "independent of"** corrected
12. **L422, there is an extra ")"**   corrected
13. **L427,   suggest changing to "and tends to be negative" and "A closer look at" or "A closer look into"**          corrected
14. **L540, change to "i.e."**          corrected

**Ozone  profile retrieval from nadir TROPOMI measurements in the UV range**

[revised manuscript text omitted]

In the future, an operational TROPOMI Ozone Profile L2 product will also be provided by ESA/KNMI (ESA/KNMI, 2021a). Due to the on-going re-calibration, this is currently  delayed and is expected to be released  after summer 2021. So far the L1B version 1 TROPOMI data  in band 3 (314 – 340 nm)  was used by Zhao et al. (2020) to determine tropospheric ozone and investigate its changing distribution due to the Covid-19 pandemy.

90 Their profile retrieval was limited to the UV3 band because of larger systematic radiance differences in band UV1 and larger fitting residuals in band UV2. The ozone profiles were derived using Optimal Estimation and a soft calibration was applied as well. Due to the narrow spectral window the vertical resolution of their retrieval is very limited (1.5 – 2 degrees of freedom).

This paper is structured as follows. Section 2 describes the data used in this paper. The TOPAS retrieval method is described in Section 3, and in Section 4  the retrieval quality based upon a sensitivity study using synthetic spectra is provided. Section

[revised manuscript text omitted]

225 The TOPAS retrieval approach is structured as follows: First, the a-priori information in the first iteration or the results from the previous iteration serve as input to the forward simulations with the radiative transfer model (RTM) in the next iteration. A polarisation correction given by a LUT is applied to the simulated intensities and will be described later in this section. In a subsequent pre-processing step, the calibration correction spectrum~and~ the rotational Raman scattering, and an offset correction are fitted to the radiance spectrum and a 1st order polynomial is subtracted. Finally, the vertical ozone profile and a

230 Lambertian (scalar) surface albedo are retrieved using Eq. 8. 7. An overview about the relevant retrieval parameters is given in Table 1.

[revised manuscript text omitted]

275 spectral window (300 – 310 nm), the scaling of the re-calibration term is not sufficient to remove the discontinuity near the channel boundary. Therefore, in this spectral window, a linear polynomial is fitted by least squares and subtracted for each fitting term separately. The third pseudo-absober accounts for a wavelength independent offset in the radiance spectra. To this end, the inverse of the solar spectrum is scaled to the radiance spectra for each of the three

280 spectral windows (Rozanov et al., 2011b).

 The Tikhonov regularisation was proven to be very efficient in dealing with nonlinear ill-posed problems  (e.g. Hasekamp and Landgraf (2001)). The Tikhonov regularisation is particularly effective in smoothing oscillations, which are typical for retrievals at a fine vertical grid.

[revised manuscript text omitted]

355 the stratosphere seems to be nearly independent  of the a-priori  profile values, as indicated by the measurement response
of about 1 (not shown). Since the vertical resolution is limited, there remains a dependence on the shape of the a-priori ozone
profile.

[revised manuscript text omitted]

In general, the vertical resolution is determined by the influence of the ozone absorption in each particular altitude region on the total strength of the ozone absorption signal registered by the instrument. Two factors are essential here and both depend on the observation/illumination geometry. These are the length of the effective light path through the altitude layers of interest and the amount of light originating from these layers. While the latter depends on the amount of the incident light and scattering properties of the atmosphere, the light paths always increase with the geometrical angles (SZA, VA, and AA). Furthermore, the simulations were done using TROPOMI specific SNR values for each viewing geometry, which also influences the sensitivity of the retrieval. The solar zenith angle (different colours) strongly affects the vertical resolution below about 17 km as less light penetrates into the lower atmosphere at large solar zenith angles. The best vertical resolution is obtained at the smallest  SZA (blue), where more light is scattered back to the instrument. The vertical resolution is also impacted by the viewing angles (plotted with different line styles)  and the azimuth angles. Below about 17 km and at large SZA, there is  a degrading vertical resolution with increasing VA  for very high AA, which is most probably related to an increased light path of the direct solar light through the troposphere thus decreasing the amount of light to be scattered. On the contrary, for lower AA the vertical resolution is improving with larger VA. The troposphere becomes invisible for the retrieval at large SZAs ($> 85°$). Depending on the viewing angle, this might be the case already at 75° SZA for large azimuth angles.

[revised manuscript text omitted]

---

## Author Comment (AC4)

**Answers to reviewer 1:**

General comments:

With this work on "Ozone Profile Retrieval from nadir TROPOMI measurements in the UV range" Mettig et al. provide an extensive first account of a Tikhonov-type retrieval of TROPOMI nadir ozone profiles and its overall performance. The research is presented clearly and exhaustively, and is of high interest to AMT. Only minor corrections and clarifications are requested, together with some improvements on internal and external referencing.

Specific comments:

**- Line 5, 142, 159, 439: Using "accuracy" as a synonym for a quantifiable total or systematic uncertainty (to be clarified by the authors) is misleading. Rather use one of the latter.**

In line 6 we are talking about uncertainty now, but in line 477 we speak about accuracy as we validate here our results using reference data with cited accuracies. In other parts of the manuscript (about MLS, OMPS-LP ...) we take the wording from the cited references and 'accuracy' is used since the numbers are derived based on validation results.

**- Line 52: Rather use "space-borne" than "at the top of the atmosphere" because the latter means different things in different applications (e.g. about 100 km for atmospheric modelling)**

It is changed in the text (Line 53).

**- Line 81: "In the future, an operational TROPOMI Ozone Profile L2 product will also be provided." Please specify by whom and/or provide a reference.**

We have added: "by ESA/KNMI" and cited the webpage
http://www.tropomi.eu/data-products/ozone-profiles (Line 82).

**- Lines 126-127: Provide a reference and possibly some interpretation of the numbers for the statement that "For the ozone profile retrieval, all ground pixels are used which do not have an error flag above 15 for "measurement_quality" or a "ground_pixel_quality" flag greater than 32."**

We added the reference to the L1B dataset documentation and gave examples for the meaning of the flags.

*"The meaning of the flag values is described in the S5P Level 1B data documentation (Kleipool,2018). For this study, pixels with error flags like "sun glint possible" or "South Atlantic Anomaly" are included." (Line 131 - 133)*

**- Line 143: Please be more specific on "Version 4.23" Does this refer to the L2 product or processor? Which L1 version does this correspond to?**

The Level 2 product version is meant. We add this in the text (Line 149).

**- Line 167: Please be somewhat more specific on the collocation criteria: "100 km maximum distance and 24 hours time difference" Are all pixels within these criteria considered, or only the 'closest' or the one with some best retrieval parameters? As TROPOMI has daily global coverage, why not take the single pixel that spatially overlaps with the sonde launch site at the day of the sonde flight, or with (the majority of) the sonde trajectory?**

We take only the closest TROPOMI pixel (if it is within the collocation criteria). Therefore, we first look for all pixels in the 100 km radius around the sonde site within 24 hour and then we take the closest pixel in time.

*"Within the 100 km radius around the ozone sonde site only the closest pixel in time is taken for the validation." (line 174-175)*

**- Lines 180-181: The term 'matching' is rather vague. It is recommended to stick to the 'collocation' terminology throughout the text.**

We changed "matching" to "collocation". (line 187)

**- Eq. (3) and following: x_a should be explained upon first use, as well as its replacement of x_0**

In our retrieval x_0 and x_a are the same. To simplify the terminology, we replaced x_0 by x_a.

**- Table 1. The "Tikhonov 0th order parameter: 11.11" is not explained in the text.**

The 0th order Tikhonov parameter is explained in line 277 *"As the zeroth order Tikhonov term, the inverse a-priori covariance matrix, $S\_a^{-1}$ is used"*.

**- Lines 236-237: "These errors are largely mitigated if the forward model is run by using the angles (viewing, solar zenith and azimuth) at the surface rather than those at the top of the atmosphere" Is this the case in this work then? Please clarify.**

Yes this is the case in this work and we clarified it by prefix "In this study we use the fact that these errors are …." in line 247.

**- Lines 240-241: "These spectra are then convolved with the TROPOMI instrument response function (ISRF)." Please provide a reference for this.**

We have added the webpage http://www.tropomi.eu/data-products/isrf-dataset as a reference. Line 253

**- Line 244: Would you have a reference for the "shift and squeeze correction"?**

We added the reference to line 257:

Rozanov, A., Bovensmann, H., Bracher, A., Hrechanyy, S., Rozanov, V., Sinnhuber, M., Stroh, F., and Burrows, J. P.: NO2 and BrO vertical profile retrieval from SCIAMACHY limb measurements: Sensitivity studies, Advances in Space Research, 36, 846–854,https://doi.org/10.1016/j.asr.2005.03.013, 2005

**- Line 255: "the Tikhonov regularisation has been proved." This statement is rather brief and hence not very clear.**

We added a reference to show how the Tikhonov regularisation is used for ozone profiles before. Together with the next sentence it should be more clear why the regularisation is used.

*" The Tikhonov regularisation was  proven to be very efficient in dealing with nonlinear ill-posed problems (e.g. Hasekamp and Landgraf, 2001). The Tikhonov regularisation is particularly effective in smoothing oscillations, which are  typical for retrievals at a fine vertical grid. (Line 272 - 274)*

**- Lines 325-326: "Consequently, the retrieval seems to be nearly independent by the a-priori." This is a quite strong statement that does not straightforwardly follow from the previous sentence. It should be quantized either here or with reference to the later analysis.**

Because it is not really a consequence, as you mentioned, we removed that from the sentence and added "in the stratosphere". Furthermore, we point out the high values of the measurement response in this range.

*"The retrieval in the stratosphere seems to be nearly independent of the a-priori profile values, as indicated by the measurement response of about 1 (not shown). Since the vertical resolution is limited, there remains a dependence on the shape of the a-priori ozone profile."* (Line 343 - 345)

**- Lines 347-350 and Figure 3 middle panel (C): The authors mix the use of 'averaging kernel' and 'averaging kernel matrix' and of their  abbreviation. Line 367 introduces "AK" as the abbreviation of "averaging kernel matrix" but later "AK" is used to indicate the individual averaging kernels. Please clearly distinguish between both for clarity, and use separate abbreviations, e.g. AK and AKM, respectively. What is shown in Figure 3 panel (C) are individual averaging kernels and not the averaging kernel matrix as a whole, so please correct the label of the horizontal axis.**

Figure 3 now shows "averaging kernel" on the X axis. In the text we now use the abbreviations "AK" for averaging kernel and use "AK matrix" for averaging kernel matrix.

**- Lines 362-363: "The noise retrieval error calculated in the linear approximation by using the Rodgers formalism" Please provide a formula of reference with formula number.**

*(Rodgers (2002),section 3.4.2, eq. 3.30) (Line 383)* is cited.

**- Figure 2: Why do the second and third rows of plots not have green shaded areas showing dispersion, as the top row for the profiles has?**

We added them to Fig 2.

**- Line 380: "The best vertical resolution is obtained at the smallest angles (blue)." Please briefly explain why this is the case.**

This is the case because the measurement there is most precise and the signal to noise ratio is the largest.

*"The best vertical resolution is obtained at the smallest SZA (blue), where more light is scattered back to the instrument." (Line 407-408)*

**- Line 384: "A measure for the mean vertical resolution…" Possibly, briefly note whether this indeed corresponds to the vertical averaging of the previous (or not exactly)?**

It is not corresponding to the averaged vertical resolution. It is a separate information which gives an idea about the information content in the retrieval. To prevent confusion, we have changed "mean vertical resolution" to " independent pieces of information".

*"A measure for independent pieces of information in the retrieved ozone profiles is provided by the number of degrees of freedom (DOF)..." (line 415)*

**- Lines 430-431: And what about latitudes beyond 50° north or south? These are shown e.g. in Figure 11, but not mentioned here.**

For the latitudes beyond 50° we use the spectral correction derived from the extratropics (-50° to -30° and 30° to 50°). This is already written in Line 468 - 470: *"The correction spectra derived from the tropical pixels (−20° to −20° latitude) are taken for latitudes up to 30° , while the spectra derived from the extra-tropics (30° to 50° latitude) are applied to all pixels above 30° latitude in both hemispheres."*

**- Figure 6, caption: Please describe the meaning of the grey area in the right plot.**

We added "*The grey area marks the 5% difference region.*" to the caption. These 5% often represent the target accuracy of stratospheric ozone profiles in the stratosphere.

**- Line 451: "convolved with the TOPAS averaging kernels" Possibly refer to Eq. (9) for clarity?**

We have added the reference to Eq. 9.

**- Lines 452-453: "The differences vanish in the altitude range where the retrieval is less sensitive." This may sound counterintuitive, so please clarify.**

We have replaced this sentence by: *"The difference between the black and red curves is particularly large in the altitude range, where  the retrieval is less sensitive (8 - 18 km)." (Line 491)*

**- Figure 8, 10, 11: Add note on different (colour) scales.**

We have added a note to the captions of Figs. 8 and 11.

*"for different altitudes (note different scales)"* and *" Each subcolumn has its own colour scale."*

**- Lines 517-519: "For most of the altitude layers, an across-track variation of the retrieved ozone number density is noticeable at some locations with higher values on the east side of the swath and lower values on the west side. This issue shall be a part of future investigations." This is not clear for all latitudes in Figure 11. Depending on the location within the orbit (especially within 40° north to south), one could say that ozone values often look artificially constant within each swath, resulting in stepwise meridian behaviour between orbits, especially for plots (B) to (D), so possibly somewhat extend this discussion.**

This comment might refer to the initial version of the manuscript. In the current version (corrections made before the paper was accepted for the discussion), this passage has been removed because Figure 11 has also been corrected.

**- Line 538: "as the precision of MLS data significantly decreases" Please provide reference.**

We added the MLS "Version 4.2x Level 2 and 3 data quality and description document" as the reference.

**- Line 576: A resolution of 9 km is mentioned in the main text (line 387).**

The vertical resolution is corrected to 6 - 9 km in the stratosphere. (line 621)

**- Lines 600-603: Refer to MLS drift / degradation studies to assess how the TROPOMI soft calibration would be affected.**

The idea of the re-calibration with MLS ozone profile for our TOPAS retrieval is that it would not be affected by any drift or degradation from MLS because it is only done once and used for all data. So far, no drift/degradation has been identified in MLS ozone profiles (Hubert et al, 2016).

**- Line 605: TROPOMI L1b, WOUDC, and SHADOZ data are obtained from third parties and should be mentioned separately.**

This has been done.

**Technical corrections:**

**Throughout: "ozonesonde(s)" can be written in one word.** corrected

**Line 51: "stratosphere" instead of "stratospheric"** corrected

**Lines 108-109: Mention abbreviation of "signal-to-noise ratio" upon first use.** corrected

**Line 133: Replace "by" by "using"** corrected

**Line 173: "laser" instead of "lasers"** corrected

**Line 217: The reference to Eq. (8) should be Eq. (7)?** corrected

**Line 220: Double period.** corrected

**Line 298: Add comma after closing bracket for readability.** corrected

**Table 2: Remove double 'with' in "European background with with polluted boundary layer"** corrected

**Line 342: Add comma after "Figure 2"** corrected

**Line 464: Double 'the'** corrected

**Line 486: "positive"** corrected

**Line 595: Replace "small" by "smaller"** corrected

**Ozone  profile retrieval from nadir TROPOMI measurements in the UV range**

[revised manuscript text omitted]

In the future, an operational TROPOMI Ozone Profile L2 product will also be provided by ESA/KNMI (ESA/KNMI, 2021a). Due to the on-going re-calibration, this is currently  delayed and is expected to be released  after summer 2021. So far, the L1B version 1 TROPOMI data  in band 3 (314 – 340 nm)  was used by Zhao et al. (2020) to determine tropospheric ozone and investigate its changing distribution due to the Covid-19 pandemy. Their profile retrieval was limited to the UV3 band because of larger systematic radiance differences in band UV1 and larger fitting residuals in band UV2. The ozone profiles were derived using Optimal Estimation and a soft calibration was applied as well. Due to the narrow spectral window the vertical resolution of their retrieval is very limited (1.5 – 2 degrees of freedom).

This paper is structured as follows. Section 2 describes the data used in this paper. The TOPAS retrieval method is described in Section 3, and in Section 4  the retrieval quality based upon a sensitivity study using synthetic spectra is provided. Section 5 discuss the additional implemented calibration correction. In Section 6, the TOPAS ozone profile retrieval is validated with  ozonesondes and lidar measurements. First results and a comparison to MLS and OMPS limb measurements are shown in Section 7. Finally, a summary is given in Section 8.

**2    Measurement data**

[revised manuscript text omitted]

225    The TOPAS retrieval approach is structured as follows: First, the a-priori information in the first iteration or the results from the previous iteration serve as input to the forward simulations with the radiative transfer model (RTM) in the next iteration. A polarisation correction given by a LUT is applied to the simulated intensities and will be described later in this section. In a subsequent pre-processing step, the calibration correction spectrum, the rotational Raman scattering, and an offset correction are fitted to the radiance spectrum and a 1st order polynomial is subtracted. Finally, the vertical ozone profile and a

230    Lambertian (scalar) surface albedo are retrieved using Eq. 7. An overview about the relevant retrieval parameters is given in Table 1.

[revised manuscript text omitted]

275 spectral window (300 – 310 nm), the scaling of the re-calibration term is not sufficient to remove the discontinuity near the channel boundary. Therefore, in this spectral window, a linear polynomial is fitted by least squares and subtracted for each fitting term separately. The third pseudo-absober accounts for a wavelength independent offset in the radiance spectra. To this end, the inverse of the solar spectrum is scaled to the radiance spectra for each of the three

280 spectral windows (Rozanov et al., 2011b).

 The Tikhonov regularisation was proven to be very efficient in dealing with nonlinear ill-posed problems  (e.g. Hasekamp and Landgraf (2001)). The Tikhonov regularisation is particularly effective in smoothing oscillations, which are typical for retrievals at a fine vertical grid.

[revised manuscript text omitted]

355  the stratosphere seems to be nearly independent  of the a-priori  profile values, as indicated by the measurement response of about 1 (not shown). Since the vertical resolution is limited, there remains a dependence on the shape of the a-priori ozone profile.

[revised manuscript text omitted]

In general, the vertical resolution is determined by the influence of the ozone absorption in each particular altitude region on the total strength of the ozone absorption signal registered by the instrument. Two factors are essential here and both depend on the observation/illumination geometry. These are the length of the effective light path through the altitude layers of interest and the amount of light originating from these layers. While the latter depends on the amount of the incident light and scattering properties of the atmosphere, the light paths always increase with the geometrical angles (SZA, VA, and AA). Furthermore, the simulations were done using TROPOMI specific SNR values for each viewing geometry, which also influences the sensitivity of the retrieval. The solar zenith angle (different colours) strongly affects the vertical resolution below about 17 km as less light penetrates into the lower atmosphere at large solar zenith angles. The best vertical resolution is obtained at the smallest  SZA (blue), where more light is scattered back to the instrument. The vertical resolution is also impacted by the viewing angles (plotted with different line styles)  and the azimuth angles. Below about 17 km and at large SZA, there is  a degrading vertical resolution with increasing VA  for very high AA, which is most probably related to an increased light path of the direct solar light through the troposphere thus decreasing the amount of light to be scattered. On the contrary, for lower AA the vertical resolution is improving with larger VA. The troposphere becomes invisible for the retrieval at large SZAs ($> 85°$). Depending on the viewing angle, this might be the case already at 75° SZA for large azimuth angles.

[revised manuscript text omitted]

---

## Author Comment (AC5)

Answers to reviewer 3:

The paper "Ozone Profile Retrieval from nadir TROPOMI measurements in the UV range" by Nora Mettig et al. provides a nice characterisation of the TOPAS algorithm, and its sensitivities to SZA, VZA, different retrieval scenarios, and shows a calibration correction.

Comments:

**- Line 42: With the last S5 launched planned in 2038, and an expected lifetime of 7 years, the nadir mission runs into the 2040s if my information is correct.**
**https://www.eumetsat.int/metop-sg**

That is correct. We have changed the date and cited the EUMETSAT webpage.

**- Line 120+121: Is the ratio between earth- and solar-spectrum actually stable enough to assume that the effects cancel out? Did you check, or is this an assumption?**

In the publication of Ludewig et al. (2020) it is assumed that "The TROPOMI instrument is designed such that all optical elements in the Earth view mode are included in the optical path when the Sun is measured. Thereby all degradation occurring in the spectrometers should cancel out when the reflectance is considered. " (Section 12, page 3571). However, it has become apparent that this might not be entirely correct because we see the need for using the re-calibration spectra, pseudo-absorber and polynomial subtraction. Nevertheless, statements in Line 120 and 121 refer to the assumption made by Ludewig et al.

(no correction made in text because Ludewig et al. is cited two sentence before and 3 sentence later)

**- Lines 126 and 127: the values '15' and '32' are not insightful to the reader. Do they refer to specific named flags, or is it a confidence scale (and if so, what is the upper limit)?**

The numbers refer to specific quality flags, which can be found in the Sentinel-5P-Level-01B-input-output-data-specification document. What we would like to express with this is that not only pixels without any error flags are used but we accept certain adverse conditions like the South Atlantic Anomaly (SAA is included in measurement_quality flag 15).

*"The meaning of the flag value is described in the S5P Level 1B data documentation (Kleipool, 2018). In this study, pixels with error flags like "sun glint possible" or "South Atlantic Anomaly" are included." (Line 131 - 133)*

**- Line 166: 24 hours time difference is really a long time for locations with gradients (i.e. winter hemispheres and the spring season in the Southern Hemisphere). At a surface wind speed of 5m/s = 432km distance in 24 hours. Wind at higher levels is much stronger. This means that you are potentially looking at a retrieval from an air mass that is not comparable to the air at the time of the sonde or lidar measurement. The air masses sampled may even be outside of your distance criterium. Can you address this issue?**

At the very beginning of our validation we manually sorted out ozone sonde profiles which definitely do not fit TROPOMI and collocated MLS profiles. There were a few profiles where we could easily see that not the same air mass was observed.

In general only a very limited number of sondes (9%) which have a time difference larger than 12 h are used. In Fig. 1 (Fig. 7 in the paper) the sonde validation plot is repeated with collocation criteria of 12 h time difference. In comparison to the original figure almost no differences are found.

[Figure]

*Figure 1: Repetition of Figure 7 from the paper with a new collocation criteria: 12 hours maximum time difference between ozone sonde and TROPOMI measurement.*

**- Line 215: Why subtract a polynomial? This is not motivated in the text (up to now).**

The first sentences of chapter 3.2 (around Line 220) should give an overview about the necessary steps in the retrieval algorithm. The motivation for the polynomial subtraction was added. Please also look at our response to your comment for Line 265 further down.

**- Line 223: To get an estimate of the potential error you introduce with scaling: is there any indication how this WFDOAS method performs with TROPOMI data in terms of accuracy? Or is this explained in Weber (2018)?**

Some discussion on the WFDOAS performance can be found in Orfanoz-Cheuquelaf et al 2021, where validation results for TROPOMI WFDOAS are presented.

We added: *"For the total ozone amount a positive bias of 2% in comparison to Brewer and Dobson instruments was found (Orfanoz-Cheuquelaf et al., 2021)."* (Line 233 - 234)

Orfanoz-Cheuquelaf, A., Rozanov, A., Weber, M., Arosio, C., Ladstätter-Weißenmayer, A., and Burrows, J. P.: Total ozone column retrieval from OMPS-NM measurements, Atmos. Meas. Tech. Discuss. [preprint], https://doi.org/10.5194/amt-2021-61, in review, 2021.

**- Line 253: It's not clear to me what the purpose of the third pseudo-absorber is. You mention that it is wavelength independent, but Irrad is not constant across the spectral range. Please clarify.**

Using the inverse solar spectrum as a pseudo-absorber in optical depth units (retrieval equation) allows us to account for a wavelength independent offset in the measured radiance within each of the three spectral windows. A constant amplitude scaling factor is fitted for each window.

*"The third pseudo-absober accounts for a wavelength independent offset in the radiance spectra. To this end, the inverse of the solar spectrum is scaled to the radiance spectra for each of the three spectral windows (Rozanov, 2011b)."* (Line 269 - 271)

**- Line 256: The subtraction of a polynomial is still not motivated. Why is this needed, and what is its effect? How are the factors of the curve established? (Or did I miss something?)**

A first order polynomial between 300 nm and 310 nm is determined by the least squares fit and subtracted for each fitting term because this wavelength region is most critical for the retrieval. As you mentioned in a later comment, there is a kind of discontinuity around 300 nm shown in Fig 5. As the scaling of the recalibration term is not sufficient to remove the discontinuity near the channel boundary, a linear polynomial is subtracted in addition. The polynomial degree (1st order) was selected after tests using different polynomial orders.

*"This pseudo-absorber, as shown in Fig 5, indicates a discontinuity around 300 nm. In the second spectral window (300 - 310 nm), the scaling of the re-calibration term is not sufficient to remove the discontinuity near the channel boundary. Therefore, in this spectral window, a linear polynomial is fitted by least squares and subtracted for each fitting term separately."* (Line 266 to 269)

**- Line 277: "Within one iterative step": Maybe you mean: "In the first iterative step..."?**

We mean "within each iterative step" and have changed the text.

**- Line 390: Somewhat more of what? Somewhat more of the shape than just the total ozone content?**

This is exactly what we mean. Theoretically, with one DOF an a priori ozone profile can only be shifted along the ozone (x-) axis and only the total column information could be retrieved. If we have more than one DOF, we can expect a change in the retrieved profile shape from the a priori.

*"This means that somewhat more information than the total ozone content can be determined in the troposphere."* (Line 419 - 420)

**- Figure 5: The discontinuity at 300 nm is quite profound. Is this due to a different binning factor, or just an inconsistent channel transition? Is the variability for wavelengths > 300 nm time dependent degradation, air mass type mis-match? Please discuss.**

For us, it is not entirely clear where these discontinuities at 300 nm come from. We don't see a clear indication of a time dependent degradation and from what we know from Ludewig et al. 2020 there should be consistency between the UV channels. It could be due to air mass mis-matches but the inconsistency seems to be greater in the tropics, but we expect more mis-matches in the extratropics because of the higher atmospheric variability.

**- Figure 5: Soft corrections based on differences between expected/simulated and measured spectra have been done before, but I think that those corrections were based on more than just one orbit in a day. Have you considered taking a full day of data to get corrections per cross track position? I think it would stabilise the effect of potential mis-matches between the simulated spectrum based on input ozone and the measured spectrum based on the real ozone distribution.**

We have considered taking a full day of TROPOMI measurements for calculating the re-calibration spectra but the effect did not seem decisive to us. In Fig. 2 (see below) we show how the tropical part of the re-calibration spectra for pixel 34 would look like for a full day (26.12.2018). There is a significant scattering especially in UV2 but it has the same order of magnitude as the scattering of the curves for single orbits on four days shown in Fig. 5 in the paper. If we average the spectra for a full day, the mean re-calibration spectra would not differ so much.

Apart from the fact that simulations for a full day are very time consuming, there is a problem with very cloudy scenes in the higher latitude region. When we tried to use a full day of data we got very irregularly distributed pixel locations in the extratropics because there are large fields of clouds. The four single orbits distributed over larger periods were carefully selected

to minimize the number of cloudy scenes.

[Figure]

*Figure 2: Re-calibration spectra (relative mean difference between measured and simulated spectra) for a full day 26.12.2018 of TROPOMI for pixel nr 34.*

**- Figure 5 & Line 420: The calibration error of 20% for small wavelengths could (in part) be caused by a lack of Rayleigh scattered light in your RTM. You may need to extend your model to 0.01hPa as Top Of Atmosphere to catch those photons in your simulation. You now seem to stop at 60km / 0.2hPa according to the conversion table in https://www.engineeringtoolbox.com/standard-atmosphere-d_604.html. I don't think that is high enough. Would it be possible to check with one (subset of an) orbit to check whether this helps to reduce the bias? If you choose to do the test and it does not change the bias curve, then just mention that in your response to my comments. If it does change the curve significantly, then act accordingly.**

As you suggested we have checked the re-calibration spectra for one orbit (and one pixel) with an extended altitude grid up to 80 km. The results are shown in Fig 3. The original re-calibration spectra used for our retrieval is given on the left in blue and the new one on the right side in green. As you correctly predicted, the large differences up to +20% and more in the lower UV1 vanish and the difference spectra for UV1 shows lower negative values in general.

[Figure]

*Figure 3: Example for re-calibration spectra with a modeled top of atmosphere 60 km (right blue) and 80 km (left green) for one orbit (5005) and detector pixel nr 34.*

To confirm our method, we repeated the evaluation of one day of TROPOMI data and the comparison with MLS for using re-calibration spectra calculated with an RTM including layers up to 80 km. Therefore we re-processed all TROPOMI cross track pixels nr 34 (UV1 counting) with the new re-calibration spectra and compared the resulting ozone profiles to MLS and the previous retrieval results. The comparison is shown in Fig 4: the previous ozone profile results in the top panels (A - C) and the new results (cut at 60 km) in the bottom panels (D - F). The comparison to MLS (B and E) and MLS x AVK (C and F) shows slightly worse results for the new re-calibration spectra between 20 km and 50 km. We found that the positive bias above 50 km is due to the lower top of atmosphere in the retrieval, but on the other hand retrieval sensitivity for altitudes above 50 km is reduced. Another interesting point is the higher vertical resolution above 45 km shown in the original retrieval results (panel A).

Using 80 km as the upper limit of the altitude grid in the RTM increases computing time because of 20% more grid points. As the soft calibration properly corrects for the missing altitudes above 60 km in the RTM, we will continue to use the soft calibration as described in the paper. The soft calibration as we have been using can then be interpreted as a combined calibration and Rayleigh scattering correction.

We have extended the description of the calibration correction our publication (and in the supplements):

*"The overall large differences in the UV1 range is a result of the top of atmosphere limited to 60~km in the RTM that leads to an underestimation of the Rayleigh scattering contribution from layers above 60~km in the RTM and retrieval. The differences between RTM calculations up to 60 and 80~km are shown in Fig. S5 in the supplements. The re-calibration spectrum obtained from RTM calculation using layers limited to 60~km can be now interpreted as a combined calibration and Rayleigh scattering correction. Figure S6 shows that this approach is very reasonable and has a positive effect on the retrieved ozone profiles." (Line 453 - 457)*

[Figure]

*Figure 4: Comparison between original retrieval reported in the manuscript (top panels) and retrieval with the re-calibration spectrum calculated with RTM layers up to 80 km (lower panels). The retrieval was processed and zonally averaged only for across track pixel nr. 34 (UV1 counting) along the orbit from October 1, 2018.*

**- Line 421: The +40% you mention, is that related to a spectral range where magnesium has absorption lines?**

That is true and we have added the sentence:

*"Such very large values correlate with the variable Magnesium absorption lines the retrieval, which is not accounted for." (Line 452)*

**- Line 431: is this a fourth pseudo absorber, on top of the 3 already mentioned earlier? What is the distribution of this scaling factor for retrievals across the Earth? Have you plotted it on a map? Does this show an expected pattern? The point I want to make is this: Using a fitted multiplication factor to explain some bias away is risky when it happens unsupervised.**

No, it is the 2nd of the three pseudo-absorbers mentioned before in Line 255. We added a note: *"(one of the three pseudo-absorbers mentioned in the previous section)" (Line ???)*

We have monitored the amplitude and scaling factors for the three pseudo-absorbers, but we haven't seen anything unexpected. In general the shift values are very small as there are no known issues with the wavelength grid. The amplitudes show the largest changes at high latitudes which can be expected because the re-calibration correction was not originally made for these latitudes. The strongest pattern in the amplitude multiplication factors are

found in the spectral window UV1 (270 - 300 nm) for the 2nd pseudo-absorber (MLS re-calibration). The amplitudes differ from 0.5 to 1.8 there. We found the highest correlation of 0.54 (Pearson factor) between the fit amplitude and the cloud fraction. This correlation seems reasonable, since clouds are not modelled in our retrieval, but are dealt with using an effective scene height. In addition, the soft calibration has been derived from cloud-free scenes only, so that larger corrections are not unexpected for cloudy scenes.

**- Line 455: "The differences vanish in the altitude range where the retrieval is less sensitive". Please be more specific. In some the altitude ranges where the retrieval is less sensitive, like the lower troposphere, the difference increases. Or is something else meant?**

We have replaced the sentence with:

*"The difference between the black and red curves is particularly large in the altitude range, where the retrieval is less sensitive (8-18 km)." (Line 491 - 492)*

**- Line 460: About the +40%: With a 1000 km colocation distance and 24h time difference, the difference in observed and reference profile can be large, especially in the UTLS. Can you tighten up those spatial and temporal colocation windows and still have a statistically relevant result, or is the dataset too limited for that?**

As shown in Fig. 1 in this document, we can tighten up the criteria to, for example, 12 hours and the results are not changing much.

**- Line 465: Would be good to re-iterate that you refer to the raw comparison, not to the AK smoothed comparison (which goes out of the 10% range).**

It is corrected in text

**- Figure 9: Why do Lauder and OHP stop near 45km? No data provided above that altitude?**

The highest altitude provided by OHP is 45 km (from the data we use). For Lauder we cut the profiles at 43 km because there was a huge scattering in the outdated dataset we used originally. Since we now use the latest dataset (given in hdf format) the scattering is reduced and now the ozone profiles up to 50 km are shown in Figs. 9 and 10.

**- Line 506: "Low ozone levels are related to cloud coverage": Are these for strictly completely cloud covered pixels? For partially covered pixels is there info below the cloud top?**

For all cloudy and partially cloudy pixels the altitude of the first retrieved layer is above the surface altitude because we calculate an effective surface altitude from the cloud top height weighted by the cloud pressure. This means, for partially cloudy pixels there is information coming from below the cloud top originating from the cloud-free part in the pixel. For completely cloudy pixels no information is coming from below the cloud top. For Fig. 11 (A) we have summed up all ozone in the subcolumn 0 km to 8 km. Since our focus is not on tropospheric ozone, we did not correct for the topography nor cloud top high. Therefore the ozone content is reduced under those conditions.

**- Line 518: Please explain what you mean with the words 'and without jumps'.**

It referred to a previously deleted passage and has now been removed.

**- Line 554: "From this... regions". How can you derive the benefit value of the TOPAS retrieval from a comparison of an a-priori (climatology) with MLS and OMPS? Whether or not you use the same a-priori climatology for TOPAS or not, if no TOPAS retrievals are used in plots I and J, then I do not see the basis for the statement in the paper. Please explain. Maybe I am missing a step in the logic.**

If the ozone a-priori climatology was perfect (±5% relative mean difference everywhere), the statement that the retrieval shows very good agreement would be relatively meaningless. A perfect retrieval might also be caused by a perfect a-priori. In Figure 12 (I) and (J) we are showing that this is not the case.

Textual comments: (all corrected)

**- Line 26: remove comma after '1980s'**   corrected

**- Line 35: remove space before 'launched'**         corrected

**- Line 74: "The main objective of this study..." I think this could be the start of a new paragraph.**    corrected

**- Line 88: LIDAR is an acronym. There are more occurences in the paper.**

We add *"light detection and ranging"* when we first mention lidar. We retain the lower case of lidar, as we believe it has now become common place.

**- Line 220: double periods after a-priori**    corrected

**- Line 220: mention of P and T profile without mentioning its source, and then at line 225 a repeat with origin from ERA-5. Please mention only once.**         corrected

**- Line 225: period after '(Hersbach et al., 2020)'**   corrected

**- Line 257: has been proved --> has been proven? (To prove, has proved, has been proven? Please check with a native English speaker).**   corrected

**- Line 293: 'is run then' --> 'is then run', or 'then runs'**    corrected

**- Line 383: 'an degrading' --> a degrading** corrected

**- Line 428: remove 'the': with time** corrected

**- Line 489: positiv needs an extra e**         corrected

**Purely for consideration:**

**- Line 21: the use of the word 'toxic'. Maybe it is a philosophical question: is ozone toxic or just very harmful when it comes into contact with other material? Personally, I would use the word toxic for substance that can cause death by ingestion into the cell material. Ozone primarily affects the surface of the skin / lungs as a reactive molecule. Since we are often not talking about internal effects of ozone inside the human cell (or plant cells), I would consider the use of the word harmful instead. I see that you have native English speakers as authors that can speak out about this. It may be that my knowledge of English nuances is too limited. I leave it up to the authors to decide.**

we change to "harmful"

**Ozone  profile retrieval from nadir TROPOMI measurements in the UV range**

[revised manuscript text omitted]

In the future, an operational TROPOMI Ozone Profile L2 product will also be provided by ESA/KNMI (ESA/KNMI, 2021a). Due to the on-going re-calibration, this is currently delayed and is expected to be released after summer 2021. So far, the L1B version 1 TROPOMI data in band 3 (314 – 340 nm) was used by Zhao et al. (2020) to determine tropospheric ozone and investigate its changing distribution due to the Covid-19 pandemy. Their profile retrieval was limited to the UV3 band because of larger systematic radiance differences in band UV1 and larger fitting residuals in band UV2. The ozone profiles were derived using Optimal Estimation and a soft calibration was applied as well. Due to the narrow spectral window the vertical resolution of their retrieval is very limited (1.5 – 2 degrees of freedom).

This paper is structured as follows. Section 2 describes the data used in this paper. The TOPAS retrieval method is described in Section 3, and in Section 4 the retrieval quality based upon a sensitivity study using synthetic spectra is provided. Section 5 discuss the additional implemented calibration correction. In Section 6, the TOPAS ozone profile retrieval is validated with ozonesondes and lidar measurements. First results and a comparison to MLS and OMPS limb measurements are shown in Section 7. Finally, a summary is given in Section 8.

**2    Measurement data**

[revised manuscript text omitted]

225      The TOPAS retrieval approach is structured as follows: First, the a-priori information in the first iteration or the results from the previous iteration serve as input to the forward simulations with the radiative transfer model (RTM) in the next iteration. A polarisation correction given by a LUT is applied to the simulated intensities and will be described later in this section. In a subsequent pre-processing step, the calibration correction spectrum, the rotational Raman scattering, and an offset correction are fitted to the radiance spectrum and a 1st order polynomial is subtracted. Finally, the vertical ozone profile and a

230 Lambertian (scalar) surface albedo are retrieved using Eq. 7. An overview about the relevant retrieval parameters is given in Table 1.

     The inversion of the ozone profile is an ill posed mathematical problem. Consequently, the retrieval needs to be constrained by a-priori information. Information on total ozone is needed, because it helps to determine the ozone profile selected from the climatology as a-priori.

235

The a-priori profiles for ozone are taken from the ozone climatology of Lamsal et al. (2004), which contains ozone profiles averaged from sonde and satellite data depending on latitude, season, and total ozone. For each processed profile, the a-priori ozone profile is scaled with an initial value for the total ozone. In order to obtain the best possible starting point, the total ozone and the effective scene albedo from the WFDOAS retrieval (Coldewey-Egbers et al., 2005; Weber et al., 2018) applied to TROPOMI is used. For the total ozone amount a positive bias of 2% in comparison to Brewer and Dobson instruments was found (?). Pressure and temperature profiles are taken from the ERA5 reanalysis (Hersbach et al., 2020). The effective scene height accounts for cloud effects and is calculated using surface height, cloud coverage and cloud-top-height. The cloud parameters are part of the operational ESA data product (Loyola et al., 2018). The cloud fraction and cloud-top-height are taken from the offline total ozone S5P product, which contains the OCRA (Optical Cloud Recognition Algorithm) cloud fraction and ROCINN_CRB (Retrieval Of Cloud Information through Neural Networks, cloud as reflecting boundary) cloud altitude matched to the UV3 channel. The cloud-top-height and surface height are weighted according to cloud coverage to determine the effective scene height (Coldewey-Egbers et al., 2005). That enables cloudy and cloudless pixels to be retrieved without a need to account for clouds implicitly in the RTM.

The radiative transfer model SCIATRAN V4.1 is used for the forward simulations (Rozanov et al., 2011a). The radiance in the wavelength range from 270 nm to 329 nm is simulated with the spectral resolution and sampling of TROPOMI. Polarisation and rotational Raman scattering are omitted for reasons of computing time and are accounted for by using a look-up table (LUT). Instead of a full-spherical atmosphere, a pseudo-spherical atmosphere is assumed, which accelerates the calculation even more. In this approximation, the direct solar beam is calculated for a fully spherical atmosphere, while for the scattered light, a plane-parallel atmosphere is assumed (Rozanov et al., 2000). This might, however, result in larger errors for larger viewing angles. These In this study we use the fact that these errors are largely mitigated if the forward model is run by using the angles (viewing, solar zenith and azimuth) at the surface rather than those at the top of the atmosphere (de Beek et al., 2004).

The radiance spectrum calculated by the RTM is corrected for polarisation using a wavelength-dependent scaling factor. To determine this factor, simulated spectra with and without polarisation are taken from the LUT with appropriate values for albedo, total ozone, geometric height, and viewing geometry. These spectra are then convolved with the TROPOMI instrument response function (ISRF) (ESA/KNMI, 2021b). The ratio of both is then used as a correction factor to account for the polarisation effects.

To account for atmospheric and instrumental effects which are not included in the forward modelling, the pre-processing fit procedure includes three pseudo-absorbers and accounts for a possible misalignment of the wavelength grids of the measured and modelled spectra by performing shift and squeeze correction (Rozanov et al., 2005). During the pre-processing step, the original wavelength range is divided into three spectral windows: 270 – 300 nm (UV1 for TROPOMI), 300 – 310 nm (lower UV2) and 310 – 329 nm (upper UV2). For each of these spectral windows, pseudo-absorbers and shift/squeeze corrections were fitted independently. The first pseudo-absorber used in the fit, which accounts for the missing contribution from the rotational Raman scattering in the forward model, is the Ring spectrum. The Ring spectrum is obtained from LUT using the same procedure as for the polarisation correction spectrum (ratio of convolved radiances modelled with and without rotational

Raman scattering contribution). The second pseudo-absorber represents the calibration correction, which accounts for errors in the stray-light correction and other systematic errors in the radiometric calibration parameters. The calibration correction spectrum is determined using the radiances modelled with ozone information from collocated MLS/Aura measurements as described in details below. This pseudo-absorber, as shown in Fig. 5, indicates a discontinuity around 300 nm. In the second spectral window (300 – 310 nm), the scaling of the re-calibration term is not sufficient to remove the discontinuity near the channel boundary. Therefore, in this spectral window, a linear polynomial is fitted by least squares and subtracted for each fitting term separately. The third pseudo-absober accounts for a wavelength independent offset in the radiance spectra and is represented by the inverse solar spectrum. In addition, a first order polynomial (linear term) is subtracted in the second spectral window (300 – 310 nm). To this end, the inverse of the solar spectrum is scaled to the radiance spectra for each of the three spectral windows (Rozanov et al., 2011b).

To cope The Tikhonov regularisation was proven to be very efficient in dealing with nonlinear ill-posed problems , the Tikhonov regularisation has been proved. Especially when a lack of stability occurs, which is indicated, for example, by oscillating ozone profiles, this type of regularisation is proposed. In the key (e.g. Hasekamp and Landgraf (2001)). The Tikhonov regularisation is particularly effective in smoothing oscillations, which are typical for retrievals at a fine vertical grid.

[revised manuscript text omitted]

355  the stratosphere seems to be nearly independent  of the a-priori  profile values, as indicated by the measurement response
of about 1 (not shown). Since the vertical resolution is limited, there remains a dependence on the shape of the a-priori ozone
profile.

[revised manuscript text omitted]

In general, the vertical resolution is determined by the influence of the ozone absorption in each particular altitude region on the total strength of the ozone absorption signal registered by the instrument. Two factors are essential here and both depend on the observation/illumination geometry. These are the length of the effective light path through the altitude layers of interest and the amount of light originating from these layers. While the latter depends on the amount of the incident light and scattering properties of the atmosphere, the light paths always increase with the geometrical angles (SZA, VA, and AA). Furthermore, the simulations were done using TROPOMI specific SNR values for each viewing geometry, which also influences the sensitivity of the retrieval. The solar zenith angle (different colours) strongly affects the vertical resolution below about 17 km as less light penetrates into the lower atmosphere at large solar zenith angles. The best vertical resolution is obtained at the smallest  SZA (blue), where more light is scattered back to the instrument. The vertical resolution is also impacted by the viewing angles (plotted with different line styles)  and the azimuth angles. Below about 17 km and at large SZA, there is  a degrading vertical resolution with increasing VA  for very high AA, which is most probably related to an increased light path of the direct solar light through the troposphere thus decreasing the amount of light to be scattered. On the contrary, for lower AA the vertical resolution is improving with larger VA. The troposphere becomes invisible for the retrieval at large SZAs ($> 85°$). Depending on the viewing angle, this might be the case already at 75° SZA for large azimuth angles.

[revised manuscript text omitted]

(panel (E)).  Such very large values correlate with the variable Magnesium absorption lines the retrieval, which is not accounted for. The overall large differences in the UV1 range is a result of the top of atmosphere limited to 60 km in the RTM

465  that leads to an underestimation of the Rayleigh scattering contribution from layers above 60 km in the RTM and retrieval. The differences between RTM calculations up to 60 and 80 km are shown in Fig. S5 in the supplements. The re-calibration spectrum obtained from RTM calculation using layers limited to 60 km can be now interpreted as a combined calibration and Rayleigh scattering correction. Figure S6 shows that this approach is very reasonable and has a positive effect on the retrieved ozone profiles. The variation of the differences between the individual orbits is very small in UV1. In UV2, there are

[revised manuscript text omitted]

In the lower troposphere ( around 5 km) an agreement with the raw ozonesonde profiles within 10% is observed in all latitude bands, which coincides with the increased vertical resolution found in  the altitude range around

[Figure]

**Figure 8.** Scatter plot of collocated TROPOMI and  ozonesonde profiles, the latter with and without convolving with TOPAS averaging kernels (red and black, respectively) for different altitudes (note different scales). The comparison between a-priori and sonde profiles is shown in grey. The solid lines show the linear fit. The R-value is indicated in the top left corner of each panel (also black and red). The ideal 1:1 curve is plotted as a dashed line.

5 km, see Fig. 4. As the difference of the  a-priori to the sonde profiles (grey) is generally larger than that for the retrieved profiles, we can  say that the retrieval improves the ozone knowledge in this altitude range for all latitude bands

520  . But we have to note that the standard deviation, in comparison to the a-priori, remains almost the same.

[revised manuscript text omitted]